

# An improved representation of aerosol acidity in the ECMWF IFS-COMPO 49R1 through the integration of EQSAM4Climv12

Samuel Rémy[1], Swen Metzger[2], Vincent Huijnen[3], Jason E Williams[3], and Johannes Flemming[4]

[1]HYGEOS, Lille, France
[2]Eco-Serve, ResearchConcepts Io GmbH, Freiburg, Germany
[3]R and D Weather and Climate modeling, Royal Netherlands Meteorological Institute, De Bilt, Netherlands
[4]European Centre for Medium Range Weather Forecasts, Reading, UK and Bonn, Germany

**Correspondence:** Samuel Rémy (sr@hygeos.com)

**Abstract.** The atmospheric composition forecasting system used to produce the CAMS forecasts of global aerosol and trace gases distributions, IFS-COMPO, undergoes periodic upgrades. In this paper we describe the development of the future operational cycle 49R1, and focus on the implementation of the thermodynamical model EQSAM4Clim version 12 for describing gas-aerosol partitioning processes for nitrate and ammonium and for providing diagnostic aerosol, cloud and precipitation pH

values at global scale. This information on aerosol acidity influences tropospheric chemistry processes associated with aqueous phase chemistry and wet deposition. The other updates to cycle 49R1 include modifications to the description of Desert Dust, Sea-salt aerosols, Carbonaceous aerosols and the size description for the calculation of aerosol optics. The implementation of EQSAM4Clim significantly improves the partitioning of reactive nitrogen compounds decreasing surface concentrations of both nitrate and ammonium, which reduces PM2.5 biases for Europe, U.S. and China, especially during summertime. For

aerosol optical depth there is generally a decrease in the simulated biases for wintertime, and for some regions an increase in the bias for summertime. Improvements in the simulated Ångström exponent is noted for almost all regions, resulting in generally a good agreement with observations. The diagnostic aerosol and precipitation pH calculated by EQSAM4Clim have been compared against results from previous simulations (for aerosol pH) and against ground observations (for precipitation pH), with the temporal distribution in the annual mean values showing good agreement against the regional observational datasets.

The use of aerosol acidity only has a relatively smaller impact on the aqueous-phase production of sulphate when compared to the changes in gas-to-particle partitioning brought by the use of EQSAM4Clim.

## 1 Introduction

The Copernicus Atmosphere Monitoring Service (CAMS) provides daily global analysis and 5-days forecasts of atmospheric composition (aerosols, trace gases and greenhouse gases) (Peuch et al. (2022)). CAMS is coordinated by the European

Centre for Medium Range Weather Forecasts (ECMWF) and uses, for its global component, the Integrated Forecasting System (IFS), with extensions to represent aerosols, trace and greenhouses gases, being called "IFS-COMPO" (also previously known as "C-IFS", Flemming et al. (2015)). In this article, we focus on the version of IFS-COMPO that simulates aerosols and trace gases. IFS-COMPO is composed of IFS(AER) for aerosols, as described in Rémy et al. (2022)



while the atmospheric chemistry is based on the chemistry module as described in Williams et al. (2022) for the tropo-
sphere and Huijnen et al. (2016) for the stratosphere. Different versions of IFS-COMPO have also been used to produce
the MACC Reanalysis (Inness et al., 2013), the CAMS interim reanalysis (Flemming et al., 2017) and the CAMS reanal-
ysis (Inness et al., 2019). Here, we primarily focus on tropospheric processes and use a version of IFS-COMPO without
the application of stratospheric chemistry. This model configuration is also referred to as IFS(CB05). Please note however
that in the operational context IFS-COMPO including stratospheric chemistry is used since the operational implementation
of cycle 48R1 on 27th of June 2023 (hereafter referred to as CY48R1). All of the components of CY48R1 IFS-COMPO
are described in detail in the publicly available IFS documentation (ECMWF, 2023), available at https://www.ecmwf.int/en/
elibrary/81374-ifs-documentation-cy48r1-part-viii-atmospheric-composition. Besides its use in CAMS, a different version of
IFS(AER) has been adopted within the Météo-France CNRM climate model system (Michou et al., 2015); an older version of
IFS-COMPO, based on cycle 43R3, is also provided as part of an OpenIFS release (Huijnen et al., 2022), while the release of
OpenIFS based on cycle 48R1 is in preparation and planned for December 2023.

IFS-COMPO has been continually updated over time, with yearly or twice yearly upgrades of the operational forecasting
system that followed and included upgrades of the operational IFS. The code revisions that are integrated into the operational
version of IFS-COMPO must satisfy the two conditions (one qualitative, one quantitative) that they bring the model closer
to "physical" reality, i.e. that more processes and/or species are represented, and that they improve the skill scores against
observations. This paper describes an upgrade of IFS-COMPO that will be implemented in cycle 49R1 (hereafter referred
to as CY49R1), aimed at a better representation of aerosol composition and gas/particle partitioning processes, through the
implementation of the thermodynamical model EQSAM4Clim (Metzger et al., 2016) to allow the representation of acidity in
the IFS-COMPO chemical forecasting system. The other aerosol modeling updates of CY49R1 are also briefly introduced.
CY49R1 is planned to become operational in the autumn 2024.

In this paper we focus on the update of the representation of acidity in the IFS-COMPO chemical forecasting system. With
increasing complexity of the representation of atmospheric aerosols in air pollution, weather and climate models, derived
properties, such as the aerosol acidity, receive more and more attention. The solution's acidity is impacted by the aerosol
composition and the water content, which itself depends on aerosol composition and meteorology. A recent literature review
of Pye et al. (2020) summarizes the current state of knowledge on the acidity of atmospheric condensed phases, specifically
particles and cloud droplets, while Tilgner et al. (2021) reviews the current state of knowledge with a focus on the acidity of
aerosol and cloud systems, which involves both inorganic and organic aqueous phase chemistry. In addition, Shah et al. (2020)
addresses the global modeling of cloud water acidity, precipitation acidity, and acid inputs to ecosystems using the GEOS-Chem
Chemistry Transport Model (CTM). Finally, Karydis et al. (2021) address how alkaline compounds control atmospheric aerosol
acidity in the EMAC model, where alkaline compounds, notably ammonium ($NH_4^+$), and to a lesser extent crustal cations e.g.
$Ca^{2+}$, which buffer aerosol pH at global scale. In the review of Tilgner et al. (2021) it is concluded that although many advances
have been made in our understanding of acidity-driven and acid-catalyzed chemical processes, many open issues still need to
be addressed for a better understanding of impact on key pollutants. More recently, a more complex representation of aerosol



acidity and its impact on multiphase chemistry has been introduced in the EC-EARTH Earth System Model, as described in Myriokefalitakis et al. (2022).

Ammonium nitrate ($NH_4NO_3$) is a key component of anthropogenic aerosols and represents a growing share of particulate matter with aerodynamical diameter below 2.5 micron (PM2.5), especially in light of the decreasing trend in sulphur dioxide ($SO_2$) emissions (Bellouin et al., 2011). In IFS-COMPO, $NH_4NO_3$ is formed through gas-particle partitioning processes, parameterized from Hauglustaine et al. (2014), while other $NO_3^-$ compounds are formed from heterogeneous reactions on desert dust and sea-salt particles (see Rémy et al. (2019) and ECMWF (2023) for more details). The implementation of the

thermodynamical model EQSAM4Clim aims to bring a more realistic representation of the production of $NH_4^+$ and $NO_3^-$ through both gas-to-particle processes and heterogeneous reactions, which in turn should improve the simulated PM2.5 and AOD, two key CAMS forecast products. EQSAM4Clim also provides an estimate of the aerosol water and aerosol acidity, which allows the diagnosis of the acidity in aerosols, clouds and precipitation. This information can then be used in the relevant multiphase chemistry processes - namely aqueous phase chemistry and wet deposition (via the solubility computation).

A better representation of gas-particle partitioning is also expected to bring an improvement in the simulated sulphur (S) and nitrogen (N) deposition terms, which are set to become new products of the CAMS portfolio because of their impact on pristine environments. Increased N deposition can cause an exceedance of critical loads in terms of the cumulative annual deposition on ecosystems (Sun et al., 2020). Elevated S and N deposition are also associated with a host of environmental issues such as acidification and eutrophication of the terrestrial system, while increasing nitrogen deposition could enhance the carbon uptake

by land processes (Reay et al., 2008).

    Briefly, the contents of this paper are as follows. In section 2 we describe the integration of EQSAM4Clim version 12 into IFS-COMPO, and how it has been coupled to other components of the system and the new pH diagnostics that have been implemented. Other atmospheric composition model updates of CY49R1 are also presented. In section 3 we compare IFS-COMPO simulations for 2019 using cycles 48R1 and pre-49R1 (i.e. including atmospheric composition model updates

planned for CY49R1, on top of CY48R1) in terms of AOD and PM2.5. We also focus on the impact of EQSAM4Clim on the oxidized and reduced N chemistry cycles, and on the subsequent impact on simulated surface concentration of oxidized and reduced nitrogen species, as well as PM2.5 and AOD. In section 4, we compare the new aerosol and precipitation pH diagnostics from IFS-COMPO against selected validation datasets: aerosol pH from ISORROPIA2 simulations using ground observations input from Zhang et al. (2021), and precipitation pH from routine monitoring networks over Europe, Asia and

U.S.

## 2   Implementation of EQSAM4Clim in IFS-COMPO and other updates of CY49R1

### 2.1   General introduction of IFS-COMPO and IFS(AER) before CY49R1

The default tropospheric chemistry scheme in IFS-COMPO, also referred to as "IFS(CB05)", is a reduced version of that developed by Yarwood et al. (2005) as described in Williams et al. (2013) and Huijnen et al. (2019), and further updated in

Williams et al. (2022) to expand on NOy species and isoprene chemistry. Photolysis is calculated online using the modified





band approach as developed in Williams et al. (2012) and embedded in IFS-COMPO, where aerosol effects are included in the calculation of photolysis rates.

The aerosol component of IFS-COMPO is a bulk aerosol scheme for all species except sea salt aerosol and desert dust, for which a sectional approach is preferred, with three bins for these two species. As such, it is often denoted as a "bulk–bin"
scheme; IFS(AER) derives from the Laboratoire d'Optique Atmosphérique/Laboratoire de Météorologie Dynamique - Zoom (LOA/LMDZ) model (Boucher et al. (2002), Reddy et al. (2005)) and uses a mass mixing ratio as the prognostic variable of the aerosol tracers. Since the implementation of operational cycle 46R1 in July 2019, the prognostic species are sea salt, desert dust, organic matter (OM), black carbon (BC), sulfate, $NO_3^-$ and $NH_4^+$. With the implementation of CY48R1 in June 2023, supplementary species have been added to explicitly represent secondary organic aerosols, with two tracers for biogenic and
anthropogenic components. IFS(AER) is by default coupled with the tropospheric chemistry of IFS-COMPO. IFS(AER) can also be run in stand-alone mode, i.e., without any interaction with the chemistry, in which case the $NO_3^-$ and $NH_4^+$ are not included and a specific tracer representing sulfur dioxide is added, as described in Rémy et al. (2019). The $NO_3^-$ production scheme (from gas-to-particle partitioning and from heterogeneous reactions) has also been adapted into the Met Office's Unified Model (UM), as described in Jones et al. (2021).

Desert dust is represented with three size bins, with radius bin limits at 0.03, 0.55, 0.9 and 20 $\mu$m. Sea salt aerosol is also represented with three size bins, with radius bin limits of 0.03, 0.5, 5 and 20 $\mu$m at 80 % relative humidity. All of the sea salt aerosol parameters (concentration, emission, deposition) are expressed at 80 % relative humidity; this is in contrast to the other aerosol species in IFS(AER), which are expressed as dry mixing ratio. The sea salt aerosol mass mixing ratio, as well as the emissions, burden and sink diagnostics, need to be divided by a factor of 4.3 to convert to dry mass mixing ratio in order to
account for the hygroscopic growth and change in particle density. There is no mass transfer between bins for either dust or sea salt.

The OM and BC aerosol species consist of their hydrophilic and hydrophobic fractions, with the ageing processes transferring mass from the hydrophobic to hydrophilic components. Sulfate aerosols (and when not fully coupled to chemistry, the precursor gas sulfur dioxide) are represented by one prognostic variable each. The $NO_3^-$ species consists of two prognostic
variables that represent fine $NO_3^-$ produced by gas–particle partitioning and coarse $NO_3^-$ produced by heterogeneous reactions of dust and sea salt particles. Finally, the secondary organics species consists of two tracers, that represent biogenic and anthropogenic Secondary Organic Aerosol (SOA). In all, the aerosol component of IFS-COMPO is thus composed of 14 prognostic variables when running stand-alone and 16 when fully coupled with chemistry (including $NO_3^-$ and $NH_4^+$), which allows for a relatively limited consumption of computing resources.

One of the most important features of the recent upgrades of IFS-COMPO is the increasing integration of aerosol and chemistry. The sulphur and nitrogen cycles are now represented across the aerosol module (for particulate species) and the chemistry module (for gaseous species). The aerosol module provides supplementary input to the chemistry in order to better represent heterogeneous reactions and the impact of aerosols on photolysis rates. The simplistic representation of the conversion of $SO_2$ into $SO_4^{2-}$ aerosol used operationally in CY45R1 and before has been replaced by a full coupling to the chemistry,
through which the $SO_4^{2-}$ production rates are computed and provided.





The production scheme of $NO_3^-$ and $NH_4^+$ through gas/particle partitioning processes, and of $NO_3^-$ from heterogeneous reactions on dust and sea salt particles, is described in Rémy et al. (2019) and has been adapted from Hauglustaine et al. (2014). It uses the first version of the EQuilibrium Simplified Aerosol Model (EQSAM, Metzger et al. (2002)). These two parameterisations use meteorological parameters as input which is provided by IFS as well as the gaseous precursors: nitric

acid ($HNO_3$) and ammonia ($NH_3$) which are provided by the chemistry module. The gas-particle partitioning scheme estimates $NO_3^-$ and $NH_4^+$ production through the neutralization of $HNO_3$ by the $NH_3$ remaining after neutralisation by sulphuric acid. The formation of $NO_3^-$ from heterogeneous reactions of $HNO_3$ with calcite (a component of dust aerosol) and sea salt particles is accounted for.

IFS-COMPO is run operationally at a horizontal resolution of $T_L 511$ (40 km grid cell), 137 levels over the vertical and a time

step of 900s. A definition of the vertical levels can be found at <https://confluence.ecmwf.int/display/UDOC/L137+model+ level+definitions>. Prognostic aerosols are used as an input of the IFS radiation scheme to compute the direct radiative effect of aerosols. IFS uses a semi-implicit semi lagrangian (SL) advection scheme (Hortal, 2002). It is computationally efficient but does not conserve the tracer mass when the flow is convergent or divergent, which is often the case in the presence of orographic features. To compensate for this, mass fixers (MF) are used for greenhouse gases (Agusti-Panareda et al., 2017), for

trace gases (Diamantakis and Flemming, 2014) and for aerosols.

## 2.2  Description of EQSAM4Clim version 12 and integration into IFS-COMPO for CY49R1

For the integration of EQSAM4Clim into IFS-COMPO we use version 12 (v12), which includes a revised calculation of aerosol acidity. This version of EQSAM4Clim is described in detail in Metzger et al. (2023); its main features are summarized below. The overall gas/liquid/solid partitioning and parameterization for calculating aerosol water uptake is identical to that described

in Metzger et al. (2016). In contrast to the original version of EQSAM (Metzger et al., 2002), EQSAM4Clim is based on a compound specific single-solute coefficient ($\nu_i$), which was introduced in Metzger et al. (2012) to accurately parameterise the single solution hygroscopic growth, considering the Kelvin effect. This $\nu_i$-approach accounts for the water uptake of concentrated nanometer-sized particles up to dilute solutions, i.e. from the compounds relative humidity of deliquescence (RHD) up to supersaturation, using the Köhler theory (Köhler, 1936). EQSAM4Clim extends the $\nu_i$-approach to multicomponent

mixtures, including semi-volatile $NH_4^+$ compounds and major crustal elements such as $Ca^{++}$, $Na^+$ and $Mg^{2+}$. A strength of EQSAM4Clim is that the entire gas–liquid–solid aerosol phase partitioning and water uptake, including major mineral cations, can be solved analytically without iterations and numerical noise, making EQSAM4Clim suited for climate and high resolution NWP applications.

EQSAM4Clim calculates the equilibrium aerosol composition and aerosol associated water mass (AW) through the neutral-

ization of anions by cations, which yields numerous salt compounds. All salt compounds (except calcium sulphate, $CaSO_4$) partition between the liquid and solid aerosol phase, depending on atmospheric temperature (T), relative humidity (RH), AW and the T-dependent RHD of (i) single solute compound solutions, or (ii) of mixed salt solutions (Metzger et al., 2012). EQSAM4Clim outputs the aerosol pH along with the concentration of aerosol species (lumped cations, anions), as well as the AW. An application of EQSAM4Clim to AW and optical depth (AOD) calculations has been presented in Metzger et al. (2018).



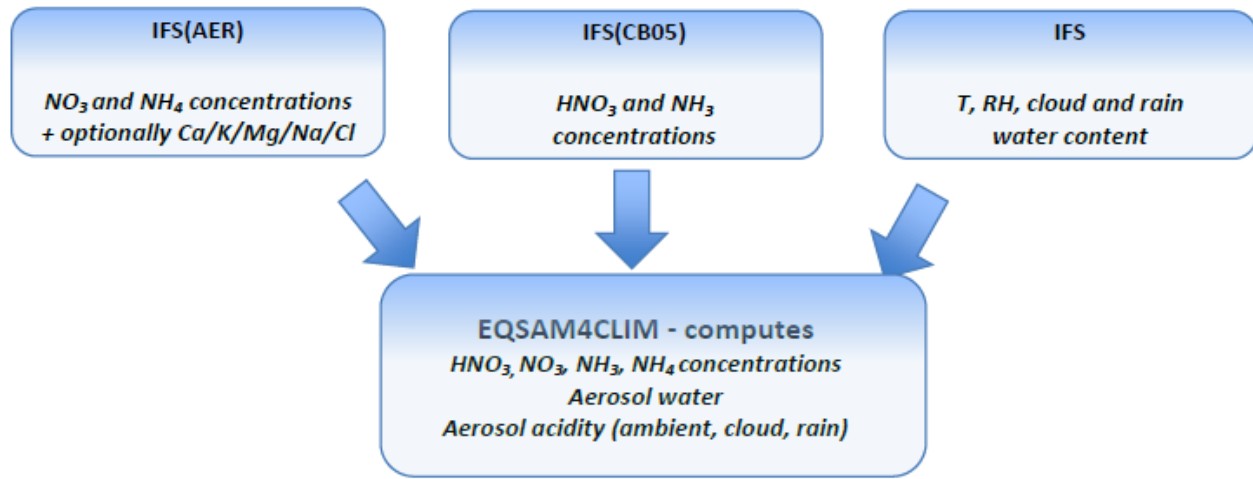

**Figure 1.** Schematic showing the inputs and outputs of EQSAM4Clim as implemented into IFS-COMPO.

A schematic of the integration of EQSAM4Clim into IFS-COMPO is shown in Figure 1. EQSAM4Clim takes as input, for each model time-step and within a given grid box, particle number and size, (i) T and RH as provided by the meteorological component of IFS-COMPO, (ii) the aerosol precursor gases, i.e., major oxidation products of natural and anthropogenic air pollution represented by $NH_3$ and $HNO_3$ from tropospheric chemistry, and (iii) the ionic aerosol concentrations lumped (liquid+solid) anions: sulphate ($SO_4^{2-}$), nitrate ($NO_3^-$), chloride ($Cl^-$), and lumped (liquid+solid) cations: ammonium ($NH_4^+$),
sodium ($Na^+$), potassium ($K^+$), magnesium ($Mg^{2+}$), and calcium ($Ca^{2+}$) as provided by IFS(AER).

It should be noted that input values for the application of EQSAM4Clim, i.e., $HNO_3/NO_3^-$, $NH_3/NH_4^+$ and $SO_4^{2-}$ are prognostic species and directly provided by IFS-COMPO, while the mineral anions ($Cl^-$) and cations ($Na^+, K^+$, $Mg^{2+}$ and $Ca^{2+}$), must be derived from the existing tracers. HCl is not yet coupled as the CB05 scheme currently does not contain tropospheric halogen chemistry. For $Ca^{2+}$, the approach chosen is the same as that described in Rémy et al. (2019): an experimental version
of IFS-COMPO that decomposes dust into a simplified mineralogical composition has been developed and is used to compute a climatology of airborne calcite. This experimental IFS-COMPO version uses as an input the dataset of soil mineralogical composition of Journet et al. (2014) to provides an estimate of the calcite content of airborne dust. Geographically dependant scaling factors derived from this climatology are used to estimate the calcium content of dust, which varies between 5-10% for fine/coarse dust and 2-5% for super-coarse dust. For $Cl^-$ and $Na^+$ input is derived from sea-salt aerosol assuming the mean
mass fractions of 55.0% $Cl^-$ and 30.6% $Na^+$ from dry sea-salt aerosol mass, following Myriokefalitakis et al. (2022). Assuming a 80% relative humidity over the ocean, this translates into 12.8% and 7.1% of the mass of sea-salt aerosol at 80% RH. The contribution of $Na^+, K^+ Mg^{2+}$ from desert dust is derived from the desert dust tracers using constant scaling factors of 1.2%, 1.5%, and 0.9%, following the approach in EC-Earth (Myriokefalitakis et al., 2022).





### 2.2.1 Computation of pH diagnostics

EQSAM4Clim calculates the pH of the solution, as governed by the $H^+$, from the electro-neutrality in solution after neutralization of all available anions by all available cations, by additionally accounting for the auto-dissociation of water, but without the dissolution and dissociation of aerosol precursor gases (e.g. $SO_2$) and weak acids (e.g. HCOOH) as this is considered in the default aqueous phase chemistry module of IFS-COMPO. Based on the RHD of the single solutes in the (mixed) solution liquid/solid partitioning is calculated, whereby all compounds for which the RH is below the RHD are assumed to be precipitated,

such that a solid and liquid phase can co-exist. The liquid-solid partitioning is strongly influenced by mineral cations and, in turn, largely determines the aerosol pH, which is also impacted by the estimated amount of AW. Because some IFS-COMPO aerosol species, such as black carbon, organic matter and secondary organics are not used in EQSAM4Clim, an amount of diagnostic aerosol water computed using relative humidity dependant growth factors for these aerosol species is used as an input to EQSAM4Clim to contribute to the estimation of aerosol acidity. More detail on how aerosol/cloud/precipitation pH is

computed can be found in Metzger et al. (2023). Please note that the domain-dependent correction factor described in Metzger et al. (2023) is not used in the implementation of EQSAM4Clim v12 in the IFS, although it is planned to include it in later IFS-COMPO upgrades.

The calculation of the cloud and rain pH of for each grid box is the cumulative contribution of aerosols, as computed by EQSAM4Clim, and of dissolved $CO_2$, methane sulphonic acid (MSA), $HSO_3^-$ and $SO_2$. In CY49R1, the $[H]^+$ contribution

from formic acid (HCOOH) and acetic acid ($CH_3COOH$) is also included.

### 2.2.2 Coupling with aqueous phase chemistry

The pH of cloud droplets affects the uptake of trace gases in cloud droplets, and hence aqueous phase chemistry in IFS(CB05). With regard to aqueous phase chemistry, the diagnostic of cloud pH is used to compute the fraction of gaseous $SO_2$ dissolved in the aqueous phase, which can subsequently be oxidised to $SO_4^{2-}$. More precisely, the cloud pH diagnostic is used to determine

the fraction of dissolved $SO_2$ which exists as $SO_2.H_2O$, bisulphite ($HSO_3^-$) and bisulphate ($SO_3^{2-}$) using pKa dissociation equilibrium values of 1.2 and 7.2, respectively. The oxidation to $SO_4^{2-}$ is then calculated as the sum of the fraction of S(IV) in the form $HSO_3^-/SO_3^{2-}$ reacting with $O_3$ and $H_2O_2$, the two principal routes for aqueous phase $SO_4^{2-}$ production using established reaction rate data (Seinfeld and Pandis, 2006). The transition metal catalysed routes for S(IV) oxidation involving iron and magnesium are neglected as no cations or anions are included in solution, where is it considered less important than the

other chemical production terms and would require a significant upgrade to the aqueous phase chemistry component, which is not the focus of this work. The subsequent change in the gas phase for $SO_2$, $NH_3$, $O_3$, $H_2O_2$, $SO_4^{2-}$ and $NH_4^+$ is then accounted for once the oxidation step has occurred.

In-cloud and below-cloud scavenging of $SO_2$ and $NH_3$ is also affected by acidity of cloud and rain water, through use of an effective Henry solubility following Seinfeld and Pandis (2006) using a fixed value of pH. Until CY47R3 this value was set to

pH of 5.5 globally. In CY48R1, this has been updated towards pH of 5.6 (over ocean) and pH of 5 (over land). In CY49R1, the diagnostic precipitation pH as provided by EQSAM4Clim is used to compute the effective Henry solubility of $SO_2$ and $NH_3$.





The impact of the use of updated cloud acidity in aqueous chemistry and rain acidity in wet deposition is found to be small in general, smaller than the impact of other contributions and in particular of the changes in gas/particle partitioning brought by the use of EQSAM4Clim. Hence, no specific discussion of this has been included, as we prefer to focus on model upgrades that have the largest impact on simulated fields.

## 2.3 Other IFS-COMPO updates included in CY49R1

In this subsection we describe additional aerosol modeling changes implemented in cycle 49R1, which also affect the results of CY49R1 model simulations presented in Section 3 as compared to the default CY48R1 configuration.

### 2.3.1 Updates to Wet deposition

In CY48R1, the aerosol and chemistry wet deposition routines of IFS-COMPO are distinct, but are both including a scheme adapted from Luo et al. (2019) which is used operationally. In order to ensure a consistent approach between aerosol and trace gases wet deposition, and to facilitate the code maintenance, the various implementations of the native chemistry and aerosol wet deposition routine have been merged into a single routine that is used to represent the wet deposition processes for both aerosols and chemical species. This new routine is called with either the chemical tracers or aerosol tracers as an input. Similar to CY48R1 and before, it is called twice with input from the large scale and convective precipitations. In the convective precipitation case, the assumed precipitation fraction has been harmonised to a value of 0.05 (0.1 was used for chemistry scavenging, 0.05 for aerosol scavenging). The following upgrades have additionally been included for aerosol wet deposition:

- Implementation of the aerosol activation parameterization of Verheggen et al. (2007). The authors provide an estimate of the fraction of aerosols that can be scavenged through in-cloud processes as a function of temperature. This parameterization is used in mixed clouds, i.e. for temperatures below the freezing point. For temperatures above 0°C, the consistency of the parameters that determine the fraction of aerosols that are subjected to in-cloud wet deposition with the results of the Verheggen parameterization has been verified.

- For below-cloud scavenging of aerosol species, the scavenging rates have been updated to follow more closely values of Croft et al. (2009), as a function of particle size. This involves updating the below cloud scavenging parameters of the table file (for rain and snow), depending on the species and the assumed size distribution, as well as implementing a below-cloud scavenging rate that is dependent on rain/snow intensity. This results in a significant increase in simulated below-cloud scavenging, which is consistent with results of Croft et al. (2009).

Both aerosol and chemistry wet deposition use the re-evaporation scheme from de Bruine et al. (2018).

### 2.3.2 Updates to Desert dust

In CY49R1, a measure of hydrophilic growth for dust has been introduced, following Chen et al. (2020), using a growth factor that increases linearly from 0.8% at 10% RH to 7.4% at 95% RH. The hydrophilic growth is used in the desert dust



optical properties, as well as in dry deposition. It is well know that desert dust are composed of mineralogical components and aggregates with very different shapes, which means that the assumed spherical shape of dust used in the Mie code (Wiscombe,

1980) that computes offline the aerosol optical properties (mass extinction, asymmetry parameter, single scattering albedo and lidar ratio) is clearly invalid. Using the online tool MOP-SMAP (Gasteiger and Wiegner, 2018), a scaling factor that accounts for the effect of asphericity has been computed, which is applied on the spherical desert dust optical properties computed with the Mie code. The assumed shape of the desert dust particles is derived from Kandler et al. (2009): a prolate spheroïd with an aspect ratio of 1.6. The resulting impact on mass extinction is significant, with an increase of 5-20% in the ultraviolet and

visible part of the spectrum for the fine and coarse bins, and of 5-10% for the super coarse bin.

### 2.3.3   Carbonaceous aerosols aging parameterization

A new parameterization of carbonaceous aerosol aging has been implemented in CY49R1, which has been adapted from Huang et al. (2013). The conversion rate from hydrophobic to hydrophilic Organic Matter (OM)/Black Carbon (BC) is represented as the sum of the conversion from oxidation and from coagulation-condensation. Oxidation corresponds to BC/OM aging through

ozone oxidation on the particle surface. Condensation is parameterized as a function of the hydroxyl (OH) radical, as described in Huang et al. (2013), while for coagulation a fixed lifetime of 20 days is adopted. This results in OM/BC aging lifetimes that range between 0.5 and 2 days at surface over most of the globe, which should be compared to the fixed value of 0.16 days used in CY48R1 and 1.16 days used in CY47R3 and before. The explicit (although simplified) representation of oxidation and condensation-coagulation processes for carbonaceous aerosols can be considered an improvement as compared to the use of

a single fixed conversion lifetime, as in CY48R1 and before. However, the impact of the new parameterization is generally small as compared to CY48R1, because the simulated aging is generally slower than the very fast fixed aging lifetime used in CY48R1 (0.16 day). As compared to CY47R3, which uses a fixed aging lifetime of 1.16 days, results are more positive in many areas.

### 2.3.4   Updates to Sea-salt aerosol

Since CY47R1, the sea-salt aerosol emissions are computed from the simulated whitecap fraction following Monahan et al. (1986), and the whitecap fraction is estimated from the surface wind speed and sea-surface temperature using the approach of Anguelova and Webster (2006), as detailed in Rémy et al. (2022). However, the Monahan et al. (1986) parameterization is applicable to particles between 0.8 and 8 $\mu m$ diameter, which means that a large fraction of the coarsest sea-salt aerosol bin is outside of the applicability range of this parameterization. Following the results of Remy and Anguelova (2021), the sea-spray

size distribution of Gong (2003) has been implemented to compute sea-salt aerosol emissions from the whitecap fraction. The computation of the whitecap fraction itself is unchanged as compared to CY47R1 and CY48R1.





### 2.3.5 Update of the PM formulae

Particulate Matter smaller than 1, 2.5 and 10 $\mu m$ are important outputs of IFS-COMPO. In CY49R1, the assumed size distribution used to compute these diagnostic outputs have been updated so as to be consistent with the one assumed in the aerosol
optics. This results in signficantly higher PM1 and PM2.5, and has a small impact on PM10. PM1, 2.5 and 10 are computed with the following formulae that uses the mass mixing ratio from each aerosol tracer as an input, denoted $[SS_{1,2,3}]$ for sea-salt aerosol, $[DD_{1,2,3}]$ for desert dust, $[NI_{1,2}]$ for nitrate, $[OM], [BC], [SU], [NI], [AM], [SOA]$ for Organic Matter, Black Carbon, Sulfate, Nitrate, Ammonium and SOA respectively :

$$PM_1 = \rho\Big(\frac{[SS_1]}{4.3} + 0.5[DD_1] + 0.96[OM] + 0.96[BC] + 0.91[SU] + 0.91[NI_1] + 0.91[AM] + 0.96[SOA]\Big)$$

$$PM_{2.5} = \rho\Big(\frac{[SS_1]}{4.3} + 0.6\frac{[SS_2]}{4.3} + [DD_1] + 0.15[DD_2] + [OM] + [BC] + [SU] + [NI_1] + 0.5[NI_2] + [AM] + [SOA]\Big)$$

$$PM_{10} = \rho\Big(\frac{[SS_1]}{4.3} + \frac{[SS_2]}{4.3} + 0.05\frac{[SS_3]}{4.3} + [DD_1] + [DD_2] + 0.4[DD_3] + [OM] + [BC] + [SU] + [NI_1] + [NI_2] + [AM] + [SOA]\Big)$$

where $\rho$ is the air density. The sea-salt aerosol tracers are divided by 4.3 so as to transform the mass mixing ratio at 80% ambient relative humidity to dry mass mixing ratio. For the hydrophilic species, aerosol water at a fixed relative humidity of
40% is also added, to account for the fact that observations are generally carried out at a fixed relative humidity of 40 to 50% RH.

### 2.3.6 Update of the aerosol optics

The aerosol optical properties (mass extinction, single scattering albedo and asymmetry parameter) are computed offline using a Mie code, as described in ECMWF (2023). The assumed size distribution used for the sulphate optical properties has been
updated, with a modal radius increase from 0.0355 to 0.11 micron, and a geometric standard deviation decrease from 2 to 1.6. Also, a 1.375 scaling factor applied to sulphate mass extinction in previous cycles, which accounted for the difference in molar mass between ammonium sulphate and sulphate has been removed, as ammonium is represented distinctly since CY46R1. This leads to a significant decrease in the sulphate mass extinction in CY49R1.

## 3 Experimental set-up and observations

### 3.1 Experiments

In this work we consider three cycling forecasts experiments without data assimilation, denoted CY48R1, CY49R1 and CY49R1_NOE4C. The experiments simulated the period from 1/12/2018 to 31/12/2019, with December 2018 used as a spin-up period. The resolution used is the current operational resolution of $T_L511$ with 137 vertical levels. CY48R1 (experiment ID b2cn) uses a configuration similar to that of the operational CY48R1, while CY49R1 (experiment ID i2dk) integrates all chem-
istry and aerosol modeling updates of IFS-COMPO intended for the operational CY49R1, including the use of EQSAM4Clim



and its coupling to tropospheric chemistry processes, but otherwise using a version of the meteorological core of IFS-COMPO identical to the CY48R1 experiment. This means that the differences between the two experiments come only from atmospheric composition modeling updates, and not from changes from the meteorological part of the IFS, or changes in the emissions. The 49R1_NOE4C experiment (experiment ID i392) deactivates the use of EQSAM4Clim, and reverts to application of the
equilibrium model available in CY48R1 (Rémy et al., 2022). All experiments use the same emissions: anthropogenic emissions from the CAMS_GLOB_ANT v5.3 dataset (Soulie et al., 2023), biomass burning emissions from the Global Fire Assimilation System version 1.4 (GFAS, Kaiser et al. (2012)), aviation emissions from CAMS-GLOB-AIR v1.1, biogenic emissions from a climatology constructed from the CAMS-GLOB-BIO v3.1 inventory and natural emissions of DMS over the ocean from CAMS-GLOB-OCE v3.1. All of the experiments use only the tropospheric chemistry component of IFS-COMPO; no explicit
stratospheric chemistry is included, as used operationally in CAMS. All simulations are run without data assimilation, so as to better assess the impact of modelling changes.

### 3.2 Observational datasets

Aerosol Optical Depth (AOD) observations are taken from the Aerosol Robotic Network (AERONET; Holben et al. (1998)) to validate the integrated column properties of the simulations. In this study AERONET level 2 data are used for evaluation, which
are cloud screened and quality assured with final calibrations, in preference to the level 1.5. AOD data at 500 nm. The $1° \times 1°$ gridded monthly merged AOD at 550nm product provided by FMI (Sogacheva et al., 2020) has also been used to evaluate simulated AOD. We also use the corresponding Angstrom exponent values derived from AERONET data integrated between 440-865nm.

Both PM2.5 and PM10 surface observations are provided by the AirNow database (https://www.airnow.gov/about-airnow/,
last access: 7 July 2023) covering U.S., the European Environment Agency (EEA) covering Europe and the China National Environmental Monitoring Center covering China. For these three regions, special care was taken to use only the rural background stations representative of locations away from strong point sources.

Observations of speciated aerosol surface concentrations from three datasets have been used: the Clean Air Status and Trends Network (CASTNET; https://www.epa.gov/castnet, last access: 27 June 2023) for the US, the European Monitoring and
Evaluation Programme (EMEP, https://ebas.nilu.no/, last access: 21 July 2023) for Europe and the Acid Deposition Monitoring Network in East Asia (EANET, https://www.eanet.asia/, last access: 12 July 2023). The CASTNET network is operated by the U.S. Environmental Protection Agency (EPA). The ambient concentrations of gases and particles are collected with an open-face three-stage filter pack at the measurement sites and a three-stage filter pack or bulk samplers at EMEP and EANET sites. For our study we choose weekly ambient concentrations of $HNO_3$, $SO_4^{2-}$, $NO_3^-$ and $NH_4^+$, which are available from 93 of the
CASTNET sites in 2019.

The Ammonia Monitoring Network (AMoN, https://nadp.slh.wisc.edu/networks/ammonia-monitoring-network/, last access on 25 July 2023) also provides three-daily observations of surface $NH_3$ for 107 sites in the U.S. . Daily observations of ambient concentrations of selected trace gases and aerosol species, including $HNO_3$, $SO_4^{2-}$, $NO_3^-$, $NH_3$ and $NH_4^+$, are available from





27 to 31 EMEP stations depending on the species. For EANET, only yearly average concentrations of $SO_4^{2-}$, $NO_3^-$ and $NH_4^+$
have been used, over 41 stations.

The datasets used to evaluate simulated aerosol and precipitation acidity are detailed in section 5, as they are quite specific.

## 4 Evaluation and impact on the simulated nitrogen life cycle

### 4.1 Comparison of chemical budgets and model fields

#### 4.1.1 Changes in global surface concentrations

Figure 2 shows the mean annual global surface concentrations for 2019 from CY48R1 for $SO_4^{2-}$, $HNO_3$, $NO_3^-$, $NH_3$ and
$NH_4^+$, along with the absolute differences when compared to the other simulations. In general the most significant changes in
precursors and particles occur between 50°N-50°S, especially over India and South-East Asia, where high fluxes of the primary
emissions of $NO_x$, $NH_3$ and $SO_2$ occur. For $SO_4^{2-}$ moderate increases occur over land, which are somewhat moderated by the
application of EQSAM4Clim. Significant increases in $HNO_3$ occur with associated decreases in $NO_3^-$ in CY49R1. Similar but
smaller changes occur for $NH_3$ and $NH_4^+$. The simulated reductions in $NH_4^+$ in CY49R1 are increases in CY49R1_NOE4C.
Thus updates to the aerosol component substantially alters the partitioning of the precursors into the particulates, which is
dominated by the use of EQSAM4Clim; the impact of the other changes in CY49R1 are smaller.

#### 4.1.2 Global chemical budgets for presursors and particles

Table 1 shows the chemical budget terms of the key species involved in the S and N cycles in IFS-COMPO for 2019 for
both gas-phase species and particulates. The $SO_2$ sources are from direct primary anthropogenic emissions with a contribution
(30%) from the oxidation of Dimethyl Sulphide by OH and outgassing/volcanic eruptions. The $SO_2$ emissions from explosive
eruptions such as that of the Raikoke on 21-25 of June 2019, are not taken into account. From Table 1 it can be seen that the
$SO_4^{2-}$ production increases by 5-6% (around 2.5 Tg yr$^{-1}$) in CY49R1_NOE4C compared to CY48R1, with a corresponding
small reduction in the global burden of $SO_2$. The tropospheric lifetime of $SO_2$ is also decreased by 10% to 2.5 days with an
associated increase in lifetime of 25% for $SO_4^{2-}$ to 4.4 days, potentially affecting AOD near strong source regions. Lifetimes
of $SO_x$ species are relatively unaffected by the application of EQSAM4Clim. The lifetime of $SO_2$ exhibits strong seasonality.
Satellite derived estimates are reported between 4-13 hours during summertime, to 24 hours during wintertime (Lee et al.,
2011);(Qu et al., 2019). For $SO_4^-$, the corresponding estimated lifetime is around 4-5 days (Chin et al., 1996);(van Noije et al.,
2014), agreeing well with our results.

For $HNO_3$ only minimal changes in the tropospheric budget occur between CY48R1 and CY49R1_NOE4C, with the dry
and wet deposition terms exhibiting only limited differences due to the updated deposition routines. With the activation of
EQSAM4Clim, the tropospheric burden of $HNO_3$ almost doubles, mainly due to a reduced fraction of $HNO_3$ being converted
into NO3_1 (from gas-to-particle partitioning). The parameterization of heterogeneous production of nitrate particles on top of
sea-salt and dust particles in EQSAM4Clim is strongly suppressed, leading to a much smaller burden of NO3_2 with CY49R1





**Table 1.** Global chemical budget terms for 2019 of $SO_2$, $SO_4^{2-}$, $NO_3^-$ aerosol from gas/particle partitioning ($NO_3\_1$) and from heterogeneous reactions ($NO_3\_2$), $HNO_3$, $NH_3$ and $NH_4^+$ as simulated by the three experiments CY48R1, CY49R1 and CY49R1_NOE4C. Fluxes are expressed in TgS yr$^{-1}$ or TgN yr$^{-1}$ and burden in TgS or TgN, respectively. For $HNO_3$, $NO_3\_1$ and $NO_3\_2$ no direct emission occurs and for $NH_3$ no chemical production term exists. Results are presented as CY48R1 / CY49R1 / CY49R1_NOE4C

| species | emissions+production | dry deposition | wet deposition | burden | lifetime (days) |
|---|---|---|---|---|---|
| $SO_2$ | 77.4 / 77.4 / 77.4 | 22.1 / 21.7 / 20.8 | 8.7 / 6.6 / 7.6 | 0.39 / 0.37 /0.35 | 2.7 / 2.5 / 2.4 |
| $SO_4$ | 46.6 / 49.2 / 49.0 | 5.5 / 5.4 / 5.3 | 41.1 / 43.8 / 43.7 | 0.44 / 0.59 / 0.59 | 3.4 / 4.4 / 4.4 |
| $HNO_3$ | 45.8 / 47.3 / 46.1 | 4.6 / 12.2 / 5.6 | 15.7 / 22.0 / 13.9 | 0.15 / 0.29 / 0.14 | 1.2 / 2.2 / 1.1 |
| $NO_3\_1$ | 3.5 / 4.7 / 2.9 | 1.0 / 0.63 / 0.79 | 2.5 / 4.1 / 2.1 | 0.05 / 0.08 / 0.04 | 4.7 / 6.2 / 4.9 |
| $NO_3\_2$ | 22 / 8.4 / 23.7 | 6.7 / 5.4 /11.4 | 15.3 / 3.0 /12.3 | 0.20 / 0.06 / 0.25 | 3.3 / 2.6 / 3.8 |
| $NH_3$ | 56.4 / 53.2 / 55.8 | 17.1 / 23.2 / 17.5 | 9.7 / 12.5 / 8.1 | 0.13 / 0.28 / 0.16 | 0.84 / 1.9 / 1.04 |
| $NH_4$ | 29.6 / 17.5 / 30.2 | 4.7 / 1.7 / 4.7 | 24.8 / 15.6 /25.5 | 0.28 / 0.19 /0.39 | 3.4 / 4.0 / 4.7 |

as compared to CY49R1_NOE4C, and contributing to the higher burden of $HNO_3$ for this experiment. Compared to CY48R1, for CY49R1_NOE4C there is only a small repartitioning of nitrate from $NO_3\_1$ towards $NO_3\_2$. This is associated to a lower $NO_3\_1$ production, caused by a higher $SO_4^{2-}$ aerosol burden, which in turn results mainly from the wet deposition updates. For CY49R1, the total burden of $NO_3^-$ decreases by around 50% compared to CY48R1. This is particularly attributed to the change in burden of $NO_3\_2$, as the burden of $NO_3\_1$ increases by more than 50%. The use of EQSAM4Clim increases the tropospheric lifetime of $NO_3\_1$ from 4.9 days up to 6.2 days, whilst reducing the lifetime of $NO_3\_2$ by 1.2 days. This essentially shifts simulated nitrate particles towards small sizes.

For the conversion of $NH_3$ to $NH_4^+$, the updates in CY49R1_NOE4C, reduces the wet deposition term for $NH_3$ by around 15% increasing both the burden and lifetime. This subsequently increases the $NH_4^+$ burden and lifetime by around 30%. The use of EQSAM4Clim in CY49R1 increase further the burden and lifetime of $NH_3$, associated with increased dry and wet deposition. Similarly to the $HNO_3/NO_3$ couple, EQSAM4Clim tilts the gas/particle conversion towards the gas phase for the $NH_3/NH_4^+$ couple, leading to substantially lower $NH_4^+$ production, burden and deposition with CY49R1 as compared to CY49R1_NOE4C, and correspondingly, a higher simulated burden and deposition of $NH_3$. This results in a lifetime of $NH_3^+$ of 1.9 days, which is significantly longer than that assumed in other studies such as (Van Damme et al., 2018), who provide a figure of 12 hours.

## 4.2 Evaluation of PM2.5

The 2019 annual mean global distribution of simulated PM2.5 are shown in Figure 3, along with the absolute differences between CY49R1_NOE4C and CY49R1 as compared against CY48R1. The highest PM2.5 concentrations occur over Africa/Middle East (associated with desert dust) and India/China (associated with anthropogenic particles), which are 3-7 times higher than the PM2.5 concentrations simulated for U.S., Europe, Russia and South America. Signatures over the ocean occur re-



**Table 2.** Regional evaluation of simulated daily PM2.5, PM10 and AOD as compared against AERONET level 2 for AOD and background rural stations for PM. The Modified Normalized Mean Bias (MNMB), Fractional Gross Error (FGE) and the Pearson correlation coefficient (R) are given, where the colours denote CY48R1 / CY49R1 / CY49R1_NOE4C. The best values are highlighted in bold. Values for the MNMB are given in $\mu g/m^3$ for PM and AOD unit for AOD.

| species | | Global | N. America | Europe | China | Africa |
|---|---|---|---|---|---|---|
| PM2.5 | MNMB | | -0.23 / **-0.03** / 0.2 | **-0.06** / 0.13 / 0.29 | -0.16 / **0.12** / 0.24 | |
| | FGE | | 0.55 / **0.51** / 0.55 | 0.5 / **0.49** / 0.52 | 0.49 / **0.47** / 0.53 | |
| | R | | 0.42 / **0.43** / 0.42 | 0.59 / **0.67** / 0.66 | 0.53 / **0.61** / 0.55 | |
| PM10 | MNMB | | | -0.10 / -0.15 / **0.** | | |
| | FGE | | | 0.46 / 0.5 / **0.45** | | |
| | R | | | **0.52** / 0.52 / 0.53 | | |
| AOD | MNMB | -0.24 / **0.04** / 0.06 | -0.28 / **0.09** / 0.14 | -0.29 / -0.06 / **0.0** | -0.15 / 0.12 / **0.11** | **0.06** / 0.14 / 0.14 |
| | FGE | 0.53 / **0.43** / 0.44 | 0.53 / **0.41** / 0.42 | 0.44 / 0.34 / **0.32** | 0.49 / **0.44** / 0.45 | 0.42 / **0.38** / **0.38** |
| | R | 0.79 / **0.80** / **0.80** | **0.74** / 0.73 / **0.74** | 0.73 / 0.76 / **0.77** | 0.72 / **0.74** / 0.72 | 0.75 / 0.78 / **0.79** |

lated to sea-salt aerosol, resulting in background PM2.5 concentrations of around 20 $\mu g/m^3$. Considering the differences of CY49R1_NOE4C compared against CY48R1, there is a moderate reduction in PM2.5 over the pristine ocean of between 2-4 $\mu g/m^3$ related to the application of an updated approach for sea-salt emissions, in contrast to an increases over land of up to 4-15 $\mu g/m^3$ over most regions. For North Africa, technical changes to the wind gusts used in the emission scheme led to a decrease in desert dust emissions, therefore reducing the PM2.5 concentrations. The largest increases occur for regions with

high anthropogenic emissions, reaching 12-18 $\mu g/m^3$ for India and East Asia, thus 25% in relative terms. These changes are reduced by the application of EQSAM4Clim, over e.g. U.S. and Europe, related to the significant reduction in the simulated surface concentration of $NH_4^+$ and $NO_3\_2$ (see Table 1).

    Figure 4 shows the weekly variability in PM2.5 as seen in the regional composites of observed and simulated particle concentrations for Europe, U.S. and China, for background rural sites. The associated Fractional Gross Error (FGE) between

simulations and observations is shown in the right column, providing some estimate as to the ability of IFS-COMPO in capturing the observations. No real seasonal cycle exists in the regional mean PM2.5 concentrations in U.S., whereas for China a strong seasonal cycle in PM2.5 exists with maximal concentrations during wintertime and, with a weaker amplitude, also in Europe. Figure 4 is complemented by Table 2 which provides the 2019 average of the Modified Normalized Mean Bias (MNMB), the FGE and the correlation coefficient of simulated PM and AOD versus observations.

For CY48R1, a strong negative bias occurs during wintertime for China and Europe, indicating muted particle formation during colder periods with shorter days, with FGE values of between 0.6-0.8. For the summertime, biases are typically lower with FGE values of between 0.3-0.4 across regions. The updates of CY49R1_NOE4C reduce the wintertime FGE to between 0.3-0.4, substantially reducing the simulated bias and improving on the representation of weekly variability in PM2.5. The





correlation between simulated and observed values is significantly higher with CY49R1 over China and Europe, as shown
in Table 2. Air quality exceedances are defined as instances where particle concentrations cross designated thresholds (e.g.
100 µg/$m^3$) for 8 hours continuously, with the observations indicating that potential occurrences are more prevalent during
wintertime. The updates made to the aerosol scheme in CY49R1_NOE4C and CY49R1 now allow IFS-COMPO to better
capture such exceedances in wintertime. In contrast, during summertime the FGE increases to 0.5-0.7 across all regions with
CY49R1_NOE4C, somewhat degrading the performance of IFS-COMPO compared to CY48R1. The use of EQSAM4Clim in
CY49R1 provides the best compromise between the other simulations, in that the wintertime improvements occur in tandem
with lower summertime biases which are similar to CY48R1 for U.S. and Europe. The largest spread between simulations
occurs for China, whose observational composite is derived from a wider range of sampling sites.

### 4.3 Evaluation of AOD

The updates of the tropospheric aerosols in IFS-COMPO CY49R1 bring significant changes in the mean simulated AOD at
550nm, as shown in Figure 5. In general, the simulated AOD shows increases in the range 0.04-0.1 AOD units in CY49R1,
except over regions which are significantly affected by desert dust. For these dry regions decreases in AOD occur of between
0-0.06 AOD units. Such decreases can be attributed to changes in the wind gust parameterization used in the dust emission
scheme, which lead to lower dust emissions in general. The largest increases in AOD occur over India and China corresponding
to the regions exhibiting significant increases in PM2.5 (c.f. Fig. 3). Over oceans, the increases in AOD are mostly caused by
the changes in the emissions of sea-salt aerosol, with the implementation of the Gong (2003) sea-spray size distribution. At
the higher latitudes the increase of the simulated AOD with the CY49R1 simulations as compared to CY48R1 is even more
significant compared to the low values, and is caused by the reduction in wet deposition in mixed and ice clouds following the
implementation of the Verheggen et al. (2007) aerosol activation parameterization in the wet deposition scheme.

Comparing the simulated AOD at 550nm by the CY48R1 and CY49R1 experiments in 2017 with the values from the FMI
merged AOD product (Sogacheva et al. (2020)) shows a relatively good agreement (Figure 6. Over oceans, the simulated
values are generally too low with CY48R1, and often too high with CY49R1; the difference is caused by updates in the sea-salt
aerosol emissions. Over regions with high anthropogenic emissions, the AOD is on average simulated too low over Europe and
India, and too high over Eastern U.S. and China with CY48R1. Over Eastern U.S., this overestimation is caused mainly by too
high biogenic secondary organics and sulphate (not shown). CY49R1 brings a small decrease in simulated AOD over China
and India, and an increase over Europe and Eastern U.S., thereby reducing bias over Europe and China, and increasing it over
Eastern U.S. and India. Figure 6 also shows the corresponding Fractional Gross Error (FGE) averaged over 2017 of monthly
simulated vs retrieved AOD at 550nm. Over most of areas, the FGE is significantly improved by CY49R1 as compared to
CY48R1. However, some degradation is noted over parts of oceans and over Eastern U.S, due to too high AOD values.

Generally, there is an increase in simulated AOD in both CY49R1_NOE4C and CY49R1 experiments, as shown by the
comparison against AERONET observations in Figure 7. Figure 7 is complemented by table 2, which also provides global
and regional skill scores (MNMB, FGE and correlation coefficient). For CY48R1 a significant low bias of -0.02 to -0.1 AOD
units occur for the global mean values and for Europe, U.S., India and China. With the exception of the U.S., the summertime





AOD bias is generally smaller. Over Africa seasonal variability is low in the observed AOD values, where no clear bias exists in CY48R1. For both CY49R1_NOE4C and CY49R1 large increases in AOD can be seen as compared to CY48R1 (between
0.25-1.0 AOD units). For more polluted regions (e.g. Europe and China) this results in a clear improvement during wintertime in the simulated bias at 500nm. For the U.S. a positive bias in AOD is introduced throughout the year and for China the positive bias enhanced during summertime. Over Africa, both CY49R1 experiments bring a small positive bias on average. The difference between CY49R1_NOE4C and CY49R1 is generally small for the AOD at 550nm, except in summertime over Europe, China and N. America, with simulated AOD at 550nm values lower by -0.02 to -0.05 AOD unit of CY49R1_NOE4C.
Overall, as indicated by Figure 7 and table 2 CY49R1 brings an improvement as compared to CY48R1 globally and for all regions in terms of FGE, and for all regions but Africa for MNMB. The impact on correlation is small but positive in general, except over North America.

## 4.4 Evaluation of the Ångström exponent

The evaluation of the weekly Ångström exponent (AE) between 440-865nm, when compared to AERONET values is shown
in Figure 8. The simulated AE decreases by around 0.1 in CY49R1_NOE4C and CY49R1 at global scale, and especially over regions in Africa, indicating that the relative amount of simulated coarse particles increases as compared to the finer particles. This results from the several updates introduced into CY49R1: the change and decrease in AOD from $SO_4^=$ , and the increase in contribution to AOD by sea-salt. With the exception of Europe, this decrease in the simulated AE leads to an improvement via lower biases as compared to CY48R1, indicating an over-estimate in the abundance of fine particles. There is a significant
change in the distribution with respect to particle size in both CY49R1_NOE4C and CY49R1. Over Europe, a negative bias of 0.2-0.3 exists in both CY49R1 and CY49R1_NOE4C. For U.S., China and India, a positive bias of 0.2-0.4 exists in CY48R1, with a reduction of between 0.1-0.2 for CY49R1_NOE4C and CY49R1. Over Africa the changes arelimited: the general decrease in simulated AE with the two CY49R1 simulations is counterbalanced by the decrease in simulated abundance of desert dust aerosols. In general, the simulated AE in CY49R1_NOE4C is quite similar to that simulated in CY49R1, showing
that the application of EQSAM4Clim only has a second-order effect on the simulated Angström exponent.

## 4.5 Evaluation of surface concentrations

Comparisons of the temporal distribution of precursor and associated particulate concentrations made against the observational surface networks show varying degrees of improvement depending on the particle type, where Table 3 provides the Mean Bias (MB), root-mean square error (RMSE) and Pearsons correlation coefficients (R) for 2019. The frequency of the measurements
varies across regions, meaning that statistics for e.g. Europe (EMEP) are more robust using daily values than for East Asia (EANET) which only provide annual values limited to a handful of stations without corresponding precursor measurements thus limiting the analysis for this region. In this subsection we focus on oxidized and reduced nitrogen species in the gas and particulate phase, because they are the species that exhibit the largest impact from the cycle 49R1 updates and the use of EQSAM4Clim in particular.



For HNO$_3$ and NO$_3^-$, density scatterplots for the comparisons against weekly aggregated CASTNET surface observations for 2019 are shown as separate panels in Figure 9 for the three simulations. For CY48R1, a small negative MB exists for HNO$_3$ in CY48R1, with a poor correlation with the observations. The associated MB for NO$_3^-$ is large and positive, again with a poor correlation (R=0.31), indicating an over-estimation in NO$_3^-$ formation. Comparing CY49R1_NOE4C shows only small reductions in the MB occur with a further decrease in R, showing that the cycle 49R1 updated other than the use of

EQSAM4Clim have a small impact for this species. The application of EQSAM4Clim in CY49R1 significantly reduces NO$_3^-$ formation (c.f. Table 1), which shifts the equilibrium towards the gaseous precursor HNO$_3$, and for U.S. subsequently increases the MB and RMSE values albeit with a much improved correlation indicating a too long lifetime under a similar production term (c.f. Table 1). For NO$_3^-$ there is a significant reduction in the simulated mean bias and error, with a corresponding large increase in correlation, which generally improves continental particle distributions and the resulting PM2.5 comparisons as

shown in Fig. 4. Therefore, the use of EQSAM4Clim in cycle 49R1 results in an improvement in the ability of IFS-COMPO towards forecasting resident NO$_3^-$ concentrations.

    The corresponding density scatter plots for the comparisons of NH$_3$ and NH$_4^+$ against AMoN/CASTNET data are shown in Figure 10. As for HNO$_3$, small negative mean bias values exist for NH$_3$ in CY48R1, with an associated weak correlation (R=0.36). For NH$_4^+$ a moderate MB exits indicating an over-estimate in particle formation similar to NO$_3^-$.. The application

of EQSAM4Clim in CY49R1 results in significant improvements in both the MB and RMSE of simulated surface NH$_4^+$ across U.S. with a general reduction in the efficacy of particle formation, as would be expected considering the improvements in the NO$_3^-$ comparisons. Again, this helps improve on any PM2.5 forecast bias in IFS-COMPO. The impact of the use of EQSAM4Clim on simulated surface concentration of NH$_3$ is smaller than the impact noticed for NH$_4^+$ but still positive, with an improvement of R from 0.36 with CY48R1 and CY49R1_NOE4C to 0.4 with CY49R1.

Figure 11 focuses on a comparison of both simulated and observed surface concentrations for both precursors and particles for a single CASTNET/AMoN measurement station, namely: Mackville in Kentucky (Lat : 37.75°N , Lon : 85.07°W). The change in the equilibrium position of gas-particle partitioning for the HNO$_3$/NO$_3^-$ couple towards less particles and more of the gas phase precursor becomes apparent when EQSAM4Clim is used, leading to a significant improvement in the representation of surface NO$_3^-$, with an associated increase in the relatively low bias simulated for HNO$_3$ in CY48R1 and CY49R1_NOE4C

to a high positive bias with CY49R1. Although the measured surface HNO$_3$ remains relatively constant throughout the year between 0.5-1$\mu g/m^3$, a strong seasonal cycle exists for surface NO$_3^-$, where colder conditions increase the particle stability under lower water vapour concentrations increasing resident concentrations two- to three-fold. This monthly variability is captured well by CY49R1. The same phenomenon occurs with to a lesser extent for the NH$_3$/NH$_4^+$ couple, again with a notable improvement in the simulated surface concentration of NH$_4^+$ by CY49R1 as compared to the other two simulations.

For NH$_3$ the observed seasonal cycle is missed across simulations, with a maxima simulated in spring and observed in summer. The timing in the simulations is strongly governed by the monthly variability and magnitude in the emission inventories, which have been show to have significant difference when compared to satellite derived emission estimates (Cao et al. (2022)). For NH$_4^+$ the excessive particle concentrations simulated in CY48R1 and CY49R1_NOE4C during summertime (+200% bias) significantly improves upon the application of EQSAM4Clim.



**Table 3.** A Regional evaluation of simulated surface concentrations of selected aerosol and trace gases species associated with particle production. The Mean Bias (MB), Root Mean Square Error (RMSE) and the Pearson correlation coefficient (R) are given, where the colours denote CY48R1 / CY49R1 / CY49R1_NOE4C. The best values are highlighted in bold. Values for the MB are given in $\mu g/m^3$. The evaluation is carried out against weekly values for CASTNET, three-daily values for AMoN, daily values for EMEP and yearly values for EANET.

| species | | U.S. (CASTNET/AMoN) | Europe (EMEP) | East Asia (EANET) |
|---|---|---|---|---|
| $SO_4$ | mean Bias | **0.41** / 0.49 / 0.65 | -0.51 / -0.42 / **-0.22** | **0.43** / 0.75 / 1.36 |
| | RMSE | 1.23 / **1.19** / 1.30 | 1.53 / 1.40 / **1.39** | **2.14** / 2.48 / 3.4 |
| | R | 0.52 / 0.52 / **0.54** | 0.21 / 0.41 / **0.45** | **0.69** / 0.66 / 0.66 |
| $HNO_3$ | mean bias | -0.11 / 0.62 / **-0.07** | -0.12 / 0.36 / **-0.10** | |
| | RMSE | **0.50** / 0.87 / 0.50 | **0.50** / 0.91 / 0.65 | |
| | R | 0.33 / **0.57** / 0.30 | **0.21** / 0.20 / 0.20 | |
| $NO_3^-$ | mean bias | 1.78 / **0.11** / 1.51 | 1.17 / **-0.08** / 0.6 | 3.19 / **-0.18** / 2.2 |
| | RMSE | 2.2 / **0.74** / 1.97 | 3.55 / **2.71** / 3.02 | 4.25 / **1.47** / 3.3 |
| | R | 0.31 / **0.65** / 0.24 | 0.19 / **0.32** / 0.21 | **0.59** / 0.5 / 0.58 |
| $NH_3$ | mean bias | -0.35 / **-0.11** / -0.33 | 0.87 / **0.82** / 0.82 | |
| | RMSE | 1.77 / **1.72** / 1.76 | 1.84 / **1.67** / 1.79 | |
| | R | 0.44 / **0.46** / 0.44 | 0.45 / **0.57** / 0.48 | |
| $NH_4^+$ | mean bias | 0.66 / **0.19** / 0.69 | 0.27 / **-0.05** / 0.33 | 1.16 / **0.63** / 1.42 |
| | RMSE | 0.90 / **0.43** / 0.92 | 0.88 / **0.69** / 0.95 | 2.03 / **1.4** / 2.5 |
| | R | 0.32 / **0.45** / 0.30 | 0.40 / **0.49** / 0.38 | 0.43 / **0.45** / 0.41 |

Table 3 summarizes the evaluation of the simulated surface concentration of selected species against observations over Europe, U.S. and East Asia. This evaluation confirms the conclusion reached using only the CASTNET/AMoN networks, showing a clear improvement for all networks of simulated surface concentration of $NO_3^-$ and $NH_4^+$ in terms of bias, RMSE and correlation coefficient. For example, the RMSE of simulated daily $NO_3^-$ as compared to EMEP observations over Europe decreases from 3.55 to 2.71 $\mu g/m^3$ by CY49R1_NOE4C as compared to CY48R1, and that of yearly $NO_3^-$ as compared

to EANET observations over East Asia from 4.25 to 1.47 $\mu g/m^3$. For $HNO_3$ and $SO_4^{2-}$, the skill scores of the CY49R1 experiments is more mixed.

## 5 Evaluation of aerosol and precipitation pH

In this section, we focus on the new diagnostic output proposed in the CY49R1 experiments: aerosol and precipitation pH. The application of EQSAM4Clim has the potential to provide an improved regional representation in the variability in both aerosol

and precipitation pH, which can act as a potential product provided by the CAMS consortium for the purposes of assessing e.g.





loads on ecosystems. To date, to our knowledge, such near-real time products on the acidity of aerosol and precipitations is not available.

## 5.1 Simulation and evaluation of aerosol pH at surface

Aerosol acidity plays an important role in a variety of aerosol physical and chemical processes as discussed in Pye et al. (2020).

In particular, it can influence production rates of secondary inorganic aerosols through heterogeneous reactions and modulate aerosol mass by controlling the gas–particle partitioning of volatile and semi-volatile compounds. Aerosol acidity is usually measured in terms of aerosol pH, which is determined by both aerosol composition and aerosol water (Zhang et al., 2021). Very few direct observations of aerosol pH are available, as it is hard to measure it directly (Li and Jang, 2012). Zhang et al. (2021) propose a data set of aerosol pH computed using ISORROPIA II (Fountoukis and Nenes, 2007) using local meteorological

variables in tandem with both gas-phase and aerosol-phase composition observations over the USA and China. In the USA, observational data are available from co-located CASTNET and AMoN sites. In China, hourly observational data from the data-sharing platform operated by the Comprehensive Observation Network for Air Pollution in Beijing-Tianjin-Hebei and Its Surrounding Areas (http://123.127.175.60:8765/siteui/index) are used. However, the Zhang et al. (2021) data set has only been calculated for the year 2011 for the U.S. and 2017 for China, while the IFS-COMPO data set is for the year 2019, which means

that the comparison between the two is only qualitative considering the change in emissions which have occurred over this decade, particularly for the U.S. dataset. We can still use such a comparison to assess the representation of regional variability.

## 5.2 Simulation and evaluation of precipitation pH

In contrast to aerosol pH, precipitation pH is measured routinely by several networks, namely the National Trends Network (NTN) operated by the National Atmospheric Deposition Program (NADP) in the U.S., EMEP in Europe and EANET in East

Asia for the regions of interest in this study, made at the same frequency as the measurements of gas-species and particles. Recently, precipitation pH has also been simulated by the GEOS-CHEM Chemistry Transport Model (CTM) for the year 2013 (Shah et al., 2020), although simulations were conducted at a coarser resolution than the IFS-COMPO simulations presented here and for a different year. In this section, the rain pH values simulated by IFS-COMPO are precipitation weighted and compared against the relevant observational data.

Figure 13 shows the average of the precipitation pH from IFS-COMPO for 2019 for CY49R1. Compared to the aerosol pH values shown in Figure 12, these values are higher (less acidic) reaching values of pH between 4.0-5.0 over oceans and 4.5-5.5 over the majority of the continents. Precipitation pH is more impacted by simulated precipitation fluxes than by aerosol composition, which is why drier areas are generally showing lower pH values, while the opposite is true for wet areas. The comparison of the precipitation pH simulation against observed values over the USA, Europe and South-East Asia reveals

that values are generally in the same range of observations, with some exceptions such as e.g. in the Mediterranean. Over the USA, the contrast between Eastern and Western U.S. is more marked in the simulated values than in the observations, although the gradient in pH values across the continent is captured. Over Europe, the simulated values agree well for most of the sites, although the simulated values are too high over Spain (dry and impacted by dust) and the Po Valley (high anthro-



pogenic emissions) indicating some additional challenges for these locations, which includes uncertainties with the simulated precipitation. For East Asia, the simulated values agree favourably for China, while they are more acidic for other countries including Indonesia.

Focusing on the US, comparisons of precipitation pH from CY49R1 against observations are shown in 14. The top two panels show time series of simulated versus observed monthly pH values at two individual sites, namely: South Pass City in Wyoming (42.47°N, 108.8°W) and Clinton in Massachusetts (42.42°N, 71.68°W), i.e., one site at the East Coast and the other at the West Coast, respectively. For South Pass City, the simulated values show a persistent negative bias of 0.2-0.4 (i.e. too acidic); however, the seasonal variability is relatively well captured. For Clinton, there is a very good agreement between the simulated and observed monthly precipitation pH, with a small negative bias of 0.1-0.2 in the spring. The bottom panel shows a density scatter-plot of a comparison of simulated monthly precipitation pH against corresponding NADP/NTN values. The agreement is much better than that for aerosol pH, albeit exhibiting a small negative bias of -0.27 pH units. This provides confidence that IFS-COMPO simulates the regional distribution and variability in precipitation pH relatively accurately, which results from a good representation of precipitation fluxes as well as aerosol composition.

Finally, Figure 15 shows a comparison of regional observed and simulated precipitation pH, including data from the GEOS-CHEM CTM as provided by Shah et al. (2020). The GEOS-CHEM CTM data is for the year 2013 instead of 2019, and using a much coarser resolution than IFS-COMPO. For all regions, the simulated values, either by IFS-COMPO or GEOS-CHEM CTM, are too low as compared to observations. The IFS-COMPO average precipitation pH values are in closer to the observed values over the U.S. and Europe, and lower than GEOS-CHEM CTM over Asia.

# 6 Conclusions

In this paper, we have presented a description and an evaluation of the model changes that will form the basis of cycle 49R1 of IFS-COMPO, focusing on the impact of the implementation and use of a simplified thermodynamical equilibrium model, EQSAM4Clim, for the purpose of improving the simulated inorganic gas/particle partitioning and the provision of aerosol/rain pH simulations. The changes bring a significant improvement in the representation of the nitrogen life cycle, mostly from the use of EQSAM4Clim, which improves the simulated biases for $NO_3^-$ and $NH_4^+$ over Europe, South-East Asia and the U.S. . This significantly improves on the simulated PM2.5 and, to a lesser extent, AOD. This provides confidence that for the future, the CAMS system will have an improved capability of simulating more accurate surface aerosol composition which is one of the important products of the service.

In addition, we have shown that the representation of aerosol and precipitation acidity in IFS-COMPO is captured with sufficient accuracy as to envisage their inclusion in the CAMS product portfolio. A preliminary evaluation of some of these potential products have been presented here, with a reasonable agreement of simulated precipitation pH against routine observations in Europe, Asia and U.S.. More evaluation is needed to fully assess the new pH diagnostics, especially concerning seasonal variability at specific locations, where this study acts as the groundwork for future datasets that could be provided by CAMS.



The other updates included in cycle 49R1 generally result in a positive impact on skill scores when applied in tandem with EQSAM4Clim. In particular, the biases in the AOD and Ångström Exponent are decreased over almost all global regions as compared to cycle 48R1. However, there are instances of an increased positive bias for e.g. AOD at 500nm and PM2.5

in spring/summertime over U.S. and China, and a persistent low bias in PM10 which require some attention. Some of the PM2.5 summertime biases are caused by too high secondary organic aerosol of biogenic origin, which currently depend on static emissions of isoprene and monoterpenes from CAMS_GLOB_BIO. A better representation of the variability of these emissions depending on the meteorological and vegetation condition could probably help in this respect, and is one direction explored for the further improvement of global CAMS products. IFS-COMPO also clearly doesn't capture some sources or

processes that lead to the formation of coarser particles as indicated by the systematic low bias in PM10. More work is needed for a better representation of PM10 in IFS-COMPO.

*Code and data availability.* Model codes developed at ECMWF are the intellectual property of ECMWF and its member states, and therefore the IFS code is not publicly available. ECMWF member-state weather services and their approved partners will get access granted. Access to an open version of the IFS code (OpenIFS) that includes cycle CY43R3 IFS(AER) ((Huijnen et al., 2022)) may be obtained from ECMWF

under an OpenIFS licence. More details at https://confluence.ecmwf.int/display/OIFS/About+OpenIFS. The EQSAM4Clim version 12 code can be found at https://doi.org/10.5281/zenodo.10276178

*Author contributions.* SR, SM and VH designed and carried out the numerical experiments used in this article. SR, SM, VH and JW drafted the paper; SM did the pH literature review, developed and implemented the revised pH formulation of EQSAM4clim into IFS; SR, VH, SM, JW and JF maintain and carry out general aerosol and trace gas developments on IFS-COMPO; all contributed to drafting and revising this

article.

*Competing interests.* At least one of the (co-)authors is a member of the editorial board of Geoscientific Model Development.

*Acknowledgements.* This work is supported by the Copernicus Atmospheric Monitoring Services (CAMS) programme managed by ECMWF on behalf of the European Commission. A large number of observational datasets have been used in this work; the authors would like to

thank all the actors that created and made public these datasets: NASA (MODIS, AERONET), the European Environment Agency (airbase), the Airnow network and the United States Environmental Protection Agency (EPA for the CASTNET), the National Atmospheric Deposition Program (NADP), the Finish Meteorological Institute (FMI, merged AOD product), and the Acid Deposition Monitoring Network in East



Asia (EANET). We acknowledge the provision of Chinese PM2.5 data by Bas Mijling (KNMI) as acquired as part of the MarcoPolo-Panda project.



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





**Figure 2.** Left (panels a,d, g, j and m): Annually averaged surface concentrations of $SO_4^{2-}$, $HNO_3$, $NO_3^-$, $NH_3$ and $NH_4^+$ (from top to bottom) as simulated by CY48R1. Middle (panels b, e, h, k and n) and right (panels c, f, i, l and o), difference CY49R1 minus CY48R1 and CY49R1_NOE4C minus CY48R1.





**Figure 3.** The global mean PM2.5 distributions in μg/m$^3$ (left, panels a, b, d) as simulated in CY48R1, CY49R1 and CY49R1_NOE4C when ran in cycling forecast mode during 2019. The right column (panels c and e) shows the absolute difference in PM 2.5 distributions as compared against the CY48R1 values.





**Figure 4.** A comparison of observed and simulated weekly PM2.5 concentrations for 2019 over background rural stations in Europe, U.S. and China is shown in the left column (panels a, c, e). The corresponding Fractional Gross Error (FGE) of simulated PM2.5 against observations is shown in the right column (panels b, d, f).





**Figure 5.** 2019 global mean AOD at 550nm (panels a, b, d, left) as simulated by the CY48R1, CY49R1 and CY49R1_NOE4C experiments. Right (panels c,f), difference with CY48R1 values. The 2019 average of VIIRS AOD at 550nm is also shown top right.





**Figure 6.** Panel a (top), 2017 average of the merged AOD at 550nm product. Panels b and c, 2017 average of CY48R1 (left) and CY49R1 (right) simulated AOD at 550nm. Panels d and e, 2017 average of Fractional Gross Error (FGE) of monthly simulated vs observed AOD at 550nm for CY48R1 (left) and CY49R1 (right).





**Figure 7.** A comparison of the global (panel a, top left) and regional (other panels) weekly simulated and observed AOD (AERONET level 2) at 500nm for 2019.





**Figure 8.** A comparison of the simulated and observed weekly global (panel a, top left) and regional (other panels) Ångström exponent (440-865nm) for 2019.





**Figure 9.** Density scatterplots of the simulated vs observed weekly surface HNO$_3$ (left) and NO$_3^-$ (right) over all CASTNET stations for 2019. Units are in $\mu g/m^3$.



**Figure 10.** Density scatterplot of simulated vs observed weekly surface NH₃ (left) and NH₄⁺ (right) over all collocated CASTNET and AMoN stations (72 stations in all) for 2019. Units are in $\mu g/m^3$.





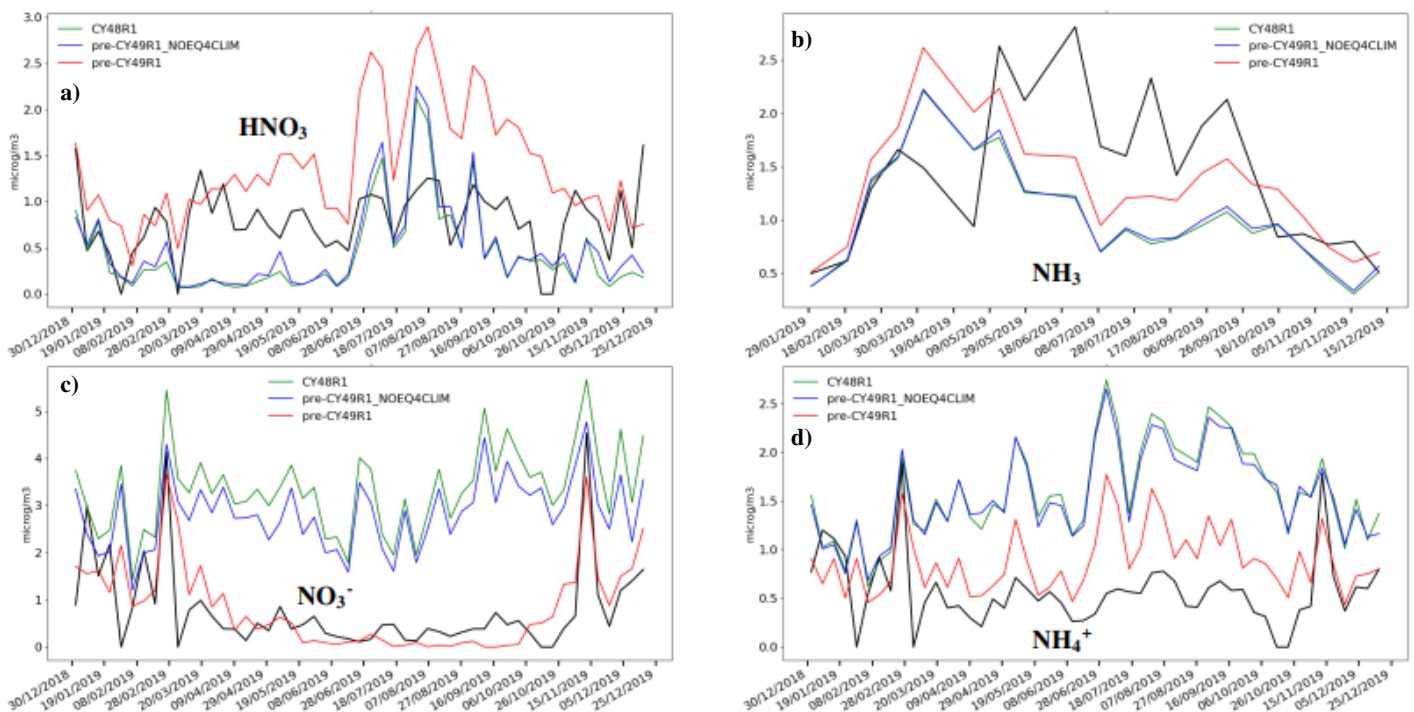

**Figure 11.** A weekly timeseries of simulated vs observed surface concentration of $HNO_3/NO_3^-$ (left, panels a and c) and $NH_3/NH_4^-$ (right, panels b and d) at the CASTNET/AMoN Mackville site (Kentucky, US).

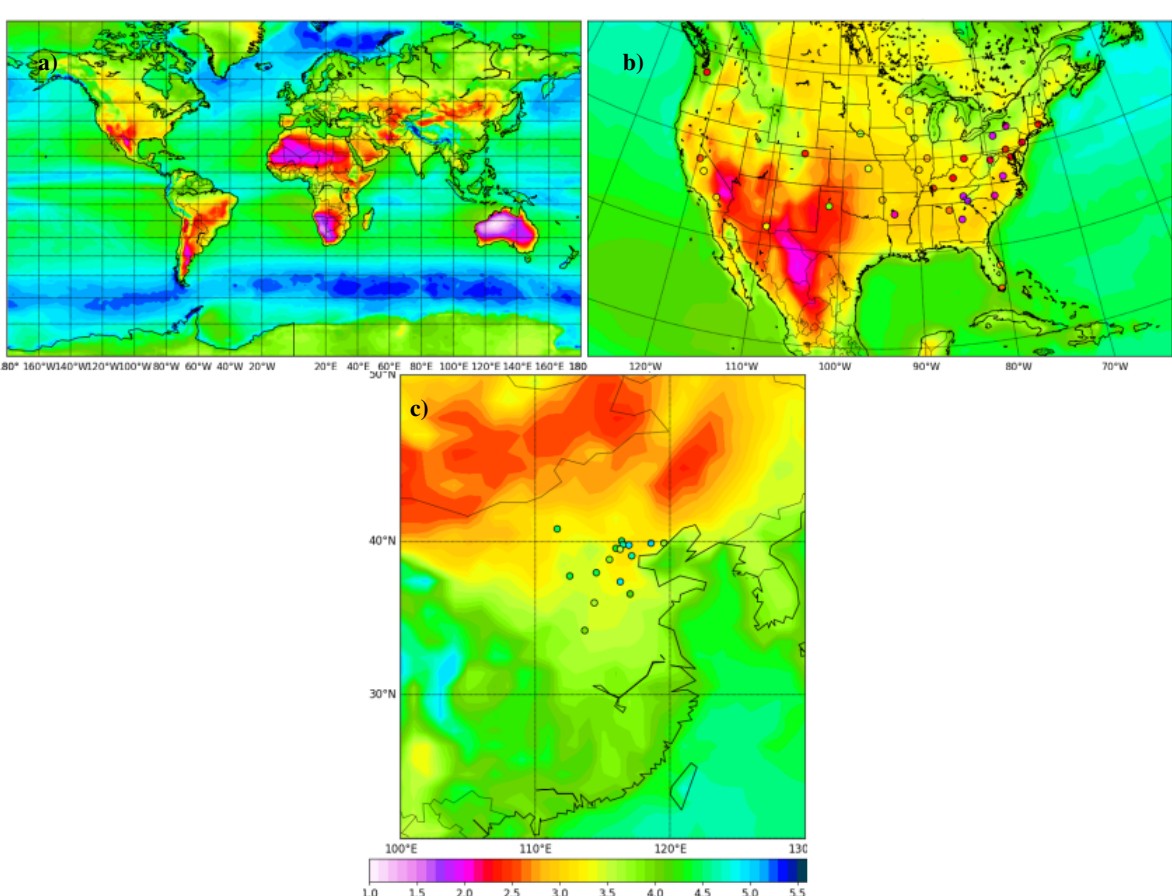

**Figure 12.** The global mean distribution of the surface aerosol pH simulated in CY49R1 (panel a, top left), along with regional comparisons against 2011 surface data from sites in the U.S. (panel b , top right) in circles taken from Zhang et al. (2021); bottom panel c, comparison against 2017 surface data from the Zhang et al. (2021) dataset over China (in circles).



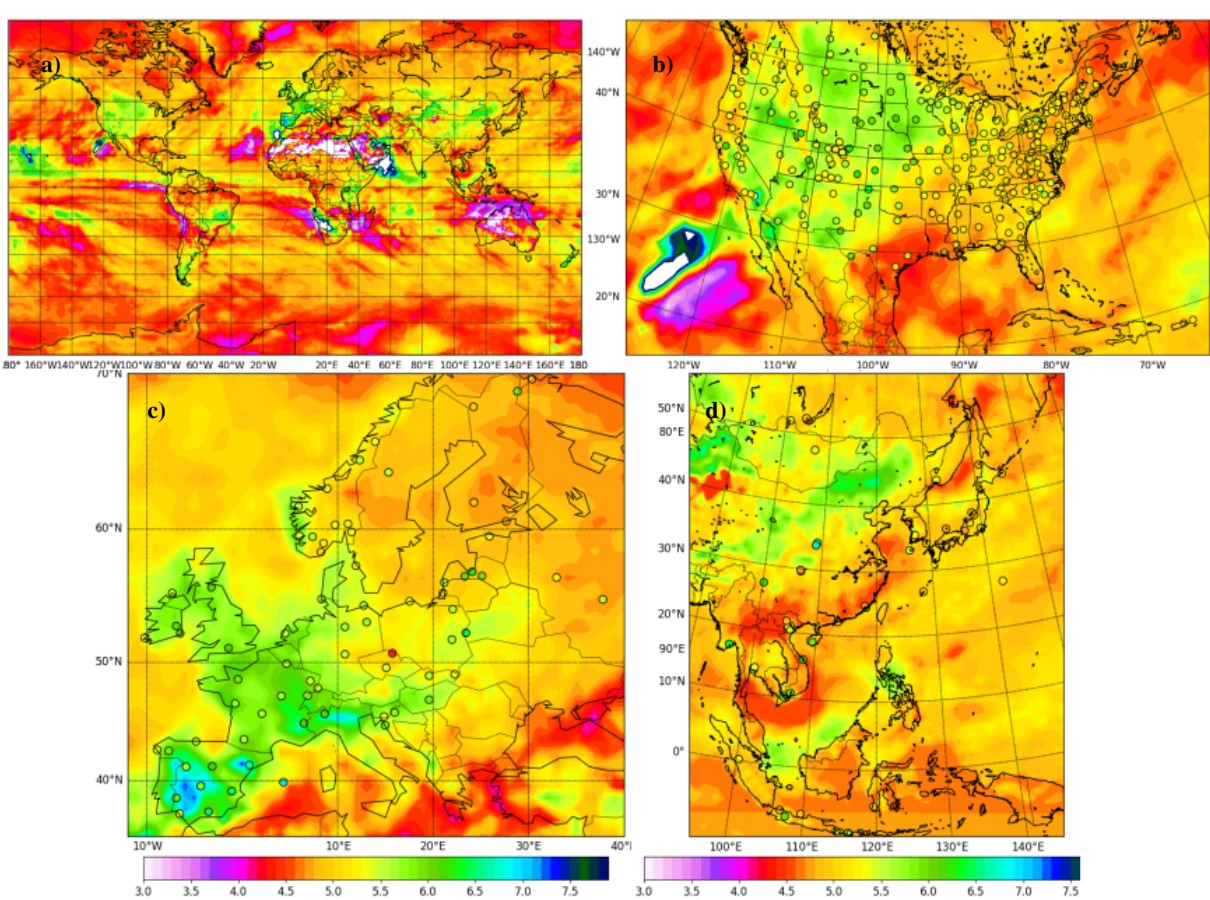

**Figure 13.** Panel a: global mean surface rain pH (top left) simulated in CY49R1 for 2019. Comparisons against the annual mean observational means for US, South-East Asia and Europe are shown in the top right, bottom left and bottom right panels (b,c,d), respectively. The respective observational mean values are shown in the circles, with the details of the regional measurements being described in the text.





**Figure 14.** A comparison of monthly averaged precipitation pH between simulated and observed values. Panels a and b show monthly comparisons for two NTN/NADP sites in the U.S: South Pass City (Wyoming) and Clinton (Massachusetts). The bottom panel c shows a density scatter plot of simulated vs observed monthly precipitation pH over all NTN/NADP stations in 2019.





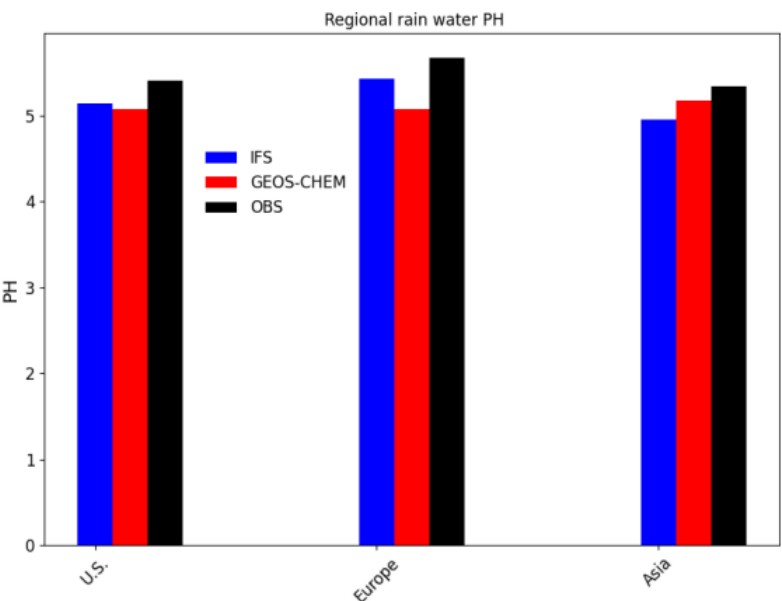

**Figure 15.** Comparison of regionally averaged rain pH as simulated by the CY49R1 experiment and observed by the NADP/NTN (U.S.), EANET (East Asia) and EMEP (Europe) networks. GEOS-CHEM simulated rain pH for the year 2013, from Shah et al. (2020), is also shown