# Peer review of "An improved representation of aerosol in the ECMWF IFS-COMPO 49R1 through the integration of EQSAM4Climv12 - a first attempt at simulating aerosol acidity"

_EGUsphere, 2023_

## Referee Comment (RC1)

**Anonymous referee comments on**

**"An improved representation of aerosol acidity in the ECMWF IFS-COMPO 49R1 through the integration of EQSAM4Climv12"**
**by Samuel Rémy et al.**

This publication presents the new version (49r1) of the IFS-COMPO model. The main new development compared to the version 48r1 is the integration of EQSAM4Clim (v12) to compute the equilibrium between gas, liquid and solid partitioning of secondary inorganic aerosol as long as crustal species. This new implementation also enables the model to deal with aerosol and precipitation pH. Wet deposition also has been updated in particular by rationalising parametrization between gas and aerosols. Other small updates have been included on desert dust, sea salt, carbonaceous aerosols and on aerosol optical properties. After presenting the changes made to the model, the authors presents an evaluation of the aerosol concentrations, especially secondary inorganic aerosols, and pH in aerosol and rain.

This publication is interesting as it presents new developments with a potential of interesting elements linked to aerosol and rain pH and the associated retroactions. The document is sometimes unclear and lacks of references. As the code is not publicly available I did not try to download and use it. Also the data correspond to the three simulations mentioned in the publication are not available.

**General comments**

The title of the publication highlights the work done on acidity in the model, but the content does not reflect this title. I would recommand change it for something like "An improved representation of aerosol in the ECMWF IFS-COMPO 49R1 through the integration of EQSAM4Climv12", and maybe add "First attempt at using aerosol acidity".

The description of the previous version of the IFS-COMPO model, cy48r1, is very unclear. I would recommand to rewrite section 2.1 by avoiding the mention of other version than 48r1, and to be more clear of how different components interacts within IFS-COMPO. Also I would recommand to add a box on Figure 1 that represents IFS-COMPO.

The newly implemented features are too rapidly described and don't give enough details to allow the reproduction of the work done. For example, it would have been interesting to have a table with old and new mass extinction for desert dust (section 2.3.2) or the details of the carbonaceous aerosols ageing parametrization (section 2.3.3).

The results section is also not so clear. I would recommand to have a section for the evaluation in which subsections refers to a comparison type (PM concentrations, AOD, etc). In each subsections you could detail the observational datasets used followed by the comparison. Also when presenting data, a good habit is to present a map with the location of the measuring points used in the study. Moreover, when evaluating simulation against gas and aerosol concentrations, it might have been interesting to also have $SO_2$ concentrations in order to evaluate the $SO_2$ oxidation. Then

another section would be used to asses the impact on simulated nitrogen and sulphur life cycle. Also I would recommand to add some possible explanations for the highlighted behaviour when possible.

Following this different reasons, I would recommand major revisions before reconsidering publication.

**Specific comments**

- Page 5, line 139: 'MF' is not used later, you can delete it

- Page 7, line 203-206: Please add a reference to support the affirmation.

- Page 8, line 212-214: Please add a reference.

- Page 10, line 277: [NI] is written twice.

- Page 10, line 284-286: Please add a reference.

- Page 10, line 299 and 304: i2dk, b2cn and i392 are not used later. Please remove them.

- Page 11, line 314: What wavelength is used for AERONET data?

- Page 11, line 317: Please keep AERONET information data together.

- Page 12, line 343: "For $SO_4^{2-}$ moderate increases occur over land" → Do you have an explanation?

- Page 12, line 344: "Which are somewhat moderated by the application of EQSAM4Clim" → Do you have an explanation?

- Page 12, line 353: "(around 2.5 Tg $yr^{-1}$)" → (2.4 Tg $yr^{-1}$)

- Page 12, line 354: "to 2.5 days" → to 2.4 days

- Page 12, line 356: "The lifetime of $SO_2$ exhibits strong seasonality" → add "not shown".

- Page 12, line 357: Lifetime you get is about 2.5 days, much greater than the ones you give as reference measured by satellite. Do you have an explanation?

- Page 12, line 358; "$SO_4^-$". There is a "2" missing.

- Page 13, line 375: $HNO_3/NO3$ → $HNO_3/NO_3^-$

- Page 13, line 377: $NH_3^+$ → $NH_3$

- Page 13, line 377-379: Do you have an explanation?

- Page 14: I would suggest to separate Table 2, into two tables, one for PM one for AOD. Also please add the number of measuring stations used for each parameters.

- Page 15, lines 405-409: Air quality exceedances are not properly defined nor shown. Please clarify the sentence and add a figure. Also I would recommand to move this part at the end of this section as it is in the middle of a coherent block.

- Page 15, line 412: "whose observational composite is derived from a wider range of sampling sites" → I don't understand this sentence, could you please clarify your thought.

- Page 15, line 417: You say that the desert dust might be modified by the changes in the wind gust parametrization. But page 11, line 302, you say that the only differences between the experiments

"come only from atmospheric composite modelling updates, and not from changes from the meteorological part of the IFS". Could please clarify the situation and illustrate the changes in the wind field if necessary.

- Page 15, line 424: "by the CY48R1 and CY49R1 experiments in 20417". Does that mean that you use other simulation for 2017 or that you compare observations for 2017 to simulations for 2019? There might a correction to do on Figure 6.

- Page 15, line 425: Figure 6 → there is a missing parenthesis.

- Page 16, line 440: between 0.25-1.00 AOD units → I don't understand to what part of the figure this part make reference.

- Page 16, line 450: "The simulated AE decreases by around 0.1 […] especially over regions in Africa." Please add a mention like 'not shown' or else as you don't provide a map of mean AE over this region.

- Page 16, line 452: "$SO_4^=$". I would recommand to have a consistent way of writing $SO_4^{2-}$.

- Page 16, line 457: "are limited". There is a missing blank.

- Page 17, line 484: "similar to $NO_3^-$." There are two points.

- Page 19, line 519-531: "Simulation and evaluation of aerosol ph at surface". In this section the authors present aerosol acidity data, but there is no exploitation of the comparison. Please move the part from the next section back to this one and elaborate the discussion.

- Page 19, section 5.2: It seems that your analyse on precipitation pH is based on the fact that precipitation amount are right. Did someone an evaluation of the precipitation rate to support this?

- Page 20, line 555: "the simulaed values show a persistent negtive bias of 0.2-0.4" → Do you have an explanation for this behaviour?

- Page 21, Acknowledgements: this section seems to need an update (MODIS, VIIRS ?, AMON, etc).

- Page 28, Figure 2: These plots are hardly readable. Maybe split it on 2 pages might help. Also, is it possible to adapt the colorbar for NH3 and NH4+ differences?

- Page 29, Figure 3: The legend on the right panels seems wrong. I think it is CY49R1-CY48R1.

- Page 30, Figure 4: Please change in the legend CY49R1NOEQSAM4CLIM to CY49R1_NOE4C. Also is it possible to add the number of stations used either in the caption or on the plots?

- Page 31, Figure 5: I think there is a mistake, "CY48R1-CY49R1" should be "CY49R1-CY48R1". Same for CY49R1_NOE4C. The caption indicates than top-right panel is VIIRS annual average, but there is no plot there.

-Page 33, Figure 7: Is it possible to add the number of stations used either in the caption or on the plots?

- Page 34, Figure 8: Same comment

- Page 35, Figure 9: Same comment. Is it possible to complete the caption with the top, middle, bottom indications, and the use of red lines?

- Page 37, Figure 11: the grey line for observations is missing in the legend.

- Page 38-39, Figure 12-13: As you compare pH data in aerosols and precipitation, it would be helpful to have the same colour scale on both figures.

- Page 40, Figure 14: top panels are blurred. Please add the number of stations for the bottom scatter plot.

- Page 41, Figure 15: Please add the number of stations for the three subdomain.

---

## Author Comment (AC2)

Response to Referee #1

*We thank the reviewer for their time and efforts reviewing this manuscript. Their comments were very helpful in improving the paper. The comments were very helpful in determining where the issues were and we have made substantive efforts to improve the clarity of the paper. In particular, we made available the simulated data on 20th of February 2024*

*The reviewer's comments are provided in regular font – our responses are shown in italics and blue. Revised sections of the manuscript are captured in quotes with starting line references in the revised manuscript included in square brackets.*

This publication presents the new version (49r1) of the IFS-COMPO model. The main new development compared to the version 48r1 is the integration of EQSAM4Clim (v12) to compute the equilibrium between gas, liquid and solid partitioning of secondary inorganic aerosol as long as crustal species. This new implementation also enables the model to deal with aerosol and precipitation pH. Wet deposition also has been updated in particular by rationalising parametrization between gas and aerosols. Other small updates have been included on desert dust, sea salt, carbonaceous aerosols and on aerosol optical properties. After presenting the changes made to the model, the authors presents an evaluation of the aerosol concentrations, especially secondary inorganic aerosols, and pH in aerosol and rain.

This publication is interesting as it presents new developments with a potential of interesting elements linked to aerosol and rain pH and the associated retroactions. The document is sometimes unclear and lacks of references. As the code is not publicly available I did not try to download and use it. Also the data correspond to the three simulations mentioned in the publication are not available.

*For the IFS code, this is unfortuntely arises from ECMWF and member's state policy. The simulation data has been made available on 20/2/2024 and can be found at* https://zenodo.org/records/10679832.

**General comments**

The title of the publication highlights the work done on acidity in the model, but the content does not reflect this title. I would recommand change it for something like "An improved representation of aerosol in the ECMWF IFS-COMPO 49R1 through the integration of EQSAM4Climv12", and maybe add "First attempt at using aerosol acidity". The description of the previous version of the IFS-COMPO model, cy48r1, is very unclear. I would recommand to rewrite section 2.1 by avoiding the mention of other version than 48r1, and to be more clear of how different components interacts within IFS-COMPO. Also I would recommand to add a box on Figure 1 that represents IFS-COMPO. The newly implemented features are too rapidly described and don't give enough details to allow the reproduction of the work done. For example, it would have been interesting to have a table with old and new mass extinction for desert dust (section 2.3.2) or the details of the carbonaceous aerosols ageing parametrization (section 2.3.3).

*Thanks for the suggestion ; the title of the article has been changed to « An improved representation of aerosol in the ECMWF IFS-COMPO 49R1 through the integration of EQSAM4Climv12 - a first attempt at simulating aerosol acidity ». Section 2.1 has been largely rewritten, and some more details have been added to the description of the new components of cycle 49R1. We would like to mention here that there when cycle 49R1 will become operational (fall 2024), an official documentation of the IFS cycle 49R1, including the atmospheric composition aspects, will be released, which will detail all aspects of the model used operationally, as was done for the cycle 48R1 documentation. In this article, we focus mainly on the implementation of EQSAM4Clim in the IFS and its impacts, and describe with relatively less detail the other updates, but we refer the reader to the future cycle 49R1 documentation, in particular for the dust specific updates. A plot has been added to show the impact of the introducing aspherical (spheroïd) particle shape for the dust tracers. A box representing IFS-COMPO has been added to Figure 1. Also, the paragraph describing the new OM aging parameterization has been deleted, as this update won't be used operationally.*

The results section is also not so clear. I would recommand to have a section for the evaluation in which subsections refers to a comparison type (PM concentrations, AOD, etc). In each subsections you could detail the observational datasets used followed by the comparison. Also when presenting data, a good habit is to present a map with the location of the measuring points used in the study. Moreover, when evaluating simulation against gas and aerosol concentrations, it might have been interesting to also have SO2 concentrations in order to evaluate the SO2 oxidation. Then another section would be used to assess the impact on simulated nitrogen and sulphur life cycle. Also I would recommand to add some possible explanations for the highlighted behaviour when possible.Following this different reasons, I would recommand major revisions before reconsidering publication.

*The sections « evaluation and impact on the simulated nitrogen life cycle » and « evaluation of aerosol and precipitation pH » have been changed and reorganized into a new « comparison of budgets and model fields » and « Evaluation » sections. The observational datasets used are now introduced at the beginning of each corresponding subsection. We use a large number of observational datasets in this work, and as a consequence it is hard to present the location of the measuring points for the observational networks used. SO2 has not been included by choice, as there are little differences between the three experiments shown : we focused on species that show a larger impact such as sulfate, nitrate, nitric acid etc. A sentence has been added :*

*« Sulfate is also shown, but SO2 is not, as there are relatively little differences between the three experiments. »*

*In the « Evaluation of surface concentrations » section. We tried to add more explanation to the many differences noted between the experiments.*

**Specific comments**

Page 5, line 139: 'MF' is not used later, you can delete it

*Removed, thank you.*

Page 7, line 203-206: Please add a reference to support the affirmation.

*We added a reference (Angle et al. 2021).*

Page 8, line 212-214: Please add a reference.

*This is meant in the context of the IFS-COMPO. The sentence has been modified to clarify this : there are no reference, as the use of aerosol acidity in chemical processes is new in the IFS.*

Page 10, line 277: [NI] is written twice.

*Corrected, thank you*

Page 10, line 284-286: Please add a reference.

*This development (including aerosol water in the PM formula) was not kept in cycle 49R1. This sentence has been deleted.*

Page 10, line 299 and 304: i2dk, b2cn and i392 are not used later. Please remove them.

*Removed, thank you.*

Page 11, line 314: What wavelength is used for AERONET data?

*500nm, the information has been added in the title and at the beginning of the « Evaluation of AOD » subsection.*

Page 11, line 317: Please keep AERONET information data together.

*Done, thank you, this paragraph (now in the « Evaluation of AOD » subsection) has also been slightly amended to mention the fact that we use the FMI merged AOD product to evaluate AOD particularly over regions where fewer AERONET stations are available.*

Page 12, line 343: "For $SO_4^{2-}$ moderate increases occur over land" → Do you have an explanation?

*This is most likely caused by the updates in wet deposition; a sentence was added at the end of this paragraph:*

*"This increase is likely associated with the reduced wet deposition in mixed clouds in the two CY49R1 simulations"*

Page 12, line 344: "Which are somewhat moderated by the application of EQSAM4Clim" → Do you have an explanation?

*This paragraph has been partly rewritten, EQSAM4Clim has little impact on the production/loss of sulfate*

Page 12, line 353: "(around 2.5 Tg yr-1)" → (2.4 Tg yr-1)

*Corrected, thank you.*

Page 12, line 354: "to 2.5 days" → to 2.4 days

*Corrected, thank you.*

Page 12, line 356: "The lifetime of SO2 exhibits strong seasonality" → add "not shown".

*Added.*

Page 12, line 357: Lifetime you get is about 2.5 days, much greater than the ones you give as reference measured by satellite. Do you have an explanation?

*The satellite values are actually between 10-20 hours in summertime to 40-60 hours in wintertime, so our simulated values are at the upper end. This is because the reported reference values correspond to surface or boundary layer values, while the values indicated in our paper correspond to global average, including areas in the upper troposphere or stratosphere where oxidation processes are much slower. The following sentence has been added:*

*"The simulated value is at the upper limit of these values, which can be explained by the fact that this is a global average over the whole atmosphere, so including areas where dry and wet oxidation processes are much slower."*

Page 12, line 358; "SO4-". There is a "2" missing.

*Corrected, thank you.*

Page 13, line 375: HNO3/NO3 → HNO3/NO3-

*Corrected, thank you.*

Page 13, line 377: NH3+ → NH3

*Corrected, thank you.*

Page 13, line 377-379: Do you have an explanation?

*Yes, the value reported in Van Damme et al. corresponds to areas closer to selected NH3 sources. Values from a global study from Luo et al. (2022), in which M. Van Damme is also involved are much closer to our simulated lifetime. The following paragraph has been added:*

*"This results in a lifetime of NH3 of 1.9 days with CY49R1, which is more than twice the value with CY48R1 (0.84 days). This is also at the high end of values found in \citet{zuo:22} from GEOS-CHEM, which range from less than 10 hours lifetime over China and Northern high latitudes up to more than 40 hours over large parts of Africa and Australia. \citep{vandamme:2018}, provides a figure of 12 hours but close to specific emission sources. The discrepancy between this short lifetime value and the CY49R1 1.9 day lifetime can be explained by different scales (global versus local) and the fact that the longer lifetimes are mostly far away from NH3 sources."*

Page 14: I would suggest to separate Table 2, into two tables, one for PM one for AOD. Also please add the number of measuring stations used for each parameters.

*The table has been split in two following your suggestion. The number of measuring stations has also been added.*

Page 15, lines 405-409: Air quality exceedances are not properly defined nor shown. Please clarify the sentence and add a figure. Also I would recommand to move this part at the end of this section as it is in the middle of a coherent block.

*The sentence has been removed. IFS is not yet evaluated in terms of air quality threshold exceedances.*

Page 15, line 412: "whose observational composite is derived from a wider range of sampling sites" → I don't understand this sentence, could you please clarify your thought.

*The sentence is indeed confusing and has been deleted. Small amendments have been brought to this paragraph.*

Page 15, line 417: You say that the desert dust might be modified by the changes in the wind gust parametrization. But page 11, line 302, you say that the only differences between the experiments "come only from atmospheric composite modelling updates, and not from changes from the meteorological part of the IFS". Could please clarify the situation and illustrate the changes in the wind field if necessary.

*The meteorological component of the IFS is indeed similar for all simulations shown here. What has changed is the input we used for the dust emission scheme: a specific gustiness was computed for that purpose, using inputs from the meteorological part. Technical changes have been brought to the gustiness diagnostic used as an input of the dust scheme, which brought a decrease of the dust emissions. The sentence has been reworded to:*

*"Such decreases can be attributed to changes in the inputs used in the dust emission scheme, which lead to lower dust emissions in general".*

Page 15, line 424: "by the CY48R1 and CY49R1 experiments in 20417". Does that mean that you use other simulation for 2017 or that you compare observations for 2017 to simulations for 2019? There might a correction to do on Figure 6.

*Yes, the three experiments have also been carried out for 2017. The following sentence has been added in the "experiments" section:*

*"Similarly, three similar experiments, which we will refer by the same names, have simulaed the period from 1/12/2016 to 31/12/2017, in order to evaluate them against observations not available in 2019"*

*Also, in the AOD section, the sentence has been clarified:*

*"Comparing the simulated AOD at 550nm by the CY48R1 and CY49R1 experiments that have simulated 2017 with the values from the FMI merged AOD product"*

Page 15, line 425: Figure 6 → there is a missing parenthesis.

*I looked but couldn't find the missing parenthesis?*

Page 16, line 440: between 0.25-1.00 AOD units → I don't understand to what part of the figure this part make reference.

*This refered to all regions, but there was a typo in the numbers. The sentence has been rewritten as:*

*"For both CY49R1\_NOE4C and CY49R1 large increases in AOD (between 0.025 and 0.1 AOD unit) can be seen as compared to CY48R1 over all regions, with a quite constant increase of 0.05 averaged globally"*

Page 16, line 450: "The simulated AE decreases by around 0.1 […] especially over regions in Africa." Please add a mention like 'not shown' or else as you don't provide a map of mean AE over this region.

*Actually over Africa, the changes in simulated AE are limited. The sentence has been rewritten as :*

*"The simulated AE decreases by around 0.1 in CY49R1\_NOE4C and CY49R1 at global scale, indicating that the relative amount of simulated coarse particles increases as compared to the finer particles."*

Page 16, line 452: "SO4=". I would recommand to have a consistent way of writing SO42-.

*Corrected, thank you.*

Page 16, line 457: "are limited". There is a missing blank.

*Corrected, thank you.*

Page 17, line 484: "similar to NO3-." There are two points.

*Corrected, thank you.*

Page 19, line 519-531: "Simulation and evaluation of aerosol ph at surface". In this section the authors present aerosol acidity data, but there is no exploitation of the comparison. Please move the part from the next section back to this one and elaborate the discussion.

*We developed a bit this part, with evaluation against the dataset gathered in Pye et al. (2020) and recent simulated values from Rosanka et al. (2024).*

Page 19, section 5.2: It seems that your analyse on precipitation pH is based on the fact that precipitation amount are right. Did someone an evaluation of the precipitation rate to support this?

*ECMWF is evaluating its precipitation forecasts from the IFS on a regular basis – it would be out of scope of this manuscript to delve into this complex subject. But precipitations is in any case on of the most complex meteorological parameters to forecast; we added a sentence to mention this:*

*"The simulation of precipitation pH depends on both aerosol composition and precipitation fluxes, and combines uncertainties from both forecasts; this is why we present here yearly and monthly values."*

*Using yearly/monthly values reduces the error from precipitation forecasts, which should be relatively accurate at a monthly/yearly scale.*

Page 20, line 555: "the simulaed values show a persistent negtive bias of 0.2-0.4" → Do you have an explanation for this behaviour?

*My interpretation is that this is a too high local source of SO2 and hence of acidic sulfate aerosols that impacts the simulated precipitation pH values here. The CY49R1 simulated surface sulfate as compared to CASTNET is much too higher over this particular area (not shown). The following sentence has been inserted into this paragraph:*

*"The negative bias is probably caused by an overestimate in simulated sulfate, itself perhaps arising from too high SO\textsubscript{2} emissions, as shown by a comparison of surface sulfate simulated by CY49R1 versus CASTNET stations in the regions (not shown)"*

Page 21, Acknowledgements: this section seems to need an update (MODIS, VIIRS ?, AMON,etc).

*We reviewed this section and added more datasets (VIIRS, Amon, the Pye et al. aerosol pH dataset).*

Page 28, Figure 2: These plots are hardly readable. Maybe split it on 2 pages might help. Also, is it possible to adapt the colorbar for NH3 and NH4+ differences?

*This Figure has been largely redrawn and split in two following your suggestion*

Page 29, Figure 3: The legend on the right panels seems wrong. I think it is CY49R1-CY48R1.- Page 30, Figure 4: Please change in the legend CY49R1NOEQSAM4CLIM to CY49R1_NOE4C. Also is it possible to add the number of stations used either in the caption or on the plots?

*Yes, thanks a lot for noticing, this has been corrected.*

Page 31, Figure 5: I think there is a mistake, "CY48R1-CY49R1" should be "CY49R1-CY48R1". Same for CY49R1_NOE4C. The caption indicates than top-right panel is VIIRS annual average, but there is no plot there.

*Similar to above, corrected, thanks a lot.*

Page 33, Figure 7: Is it possible to add the number of stations used either in the caption or on the plots?

*Done*

Page 34, Figure 8: Same comment

*Done*

Page 35, Figure 9: Same comment. Is it possible to complete the caption with the top, middle,

bottom indications, and the use of red lines?

*Done*

Page 37, Figure 11: the grey line for observations is missing in the legend.

*This has been added in the caption.*

Page 38-39, Figure 12-13: As you compare pH data in aerosols and precipitation, it would be

helpful to have the same colour scale on both figures.

*The problem is that the typical values for aerosol pH are much lower than that of precipitation pH; hence using the same color scale will blur features in both cases. The plots have been redrawn but we thought it better to use distinct colorscales between aerosol and precipitation pH.*

Page 40, Figure 14: top panels are blurred. Please add the number of stations for the bottom scatter plot.

*Done.*

Page 41, Figure 15: Please add the number of stations for the three subdomain

*Done.*

Referee #2

*We thank the reviewer for their time and efforts reviewing this manuscript. Their comments were very helpful in improving the paper. The comments were very helpful in improving the manuscript and its Figures in particular.*

*The reviewer's comments are provided in regular font – our responses are shown in italics and blue. Revised sections of the manuscript are captured in quotes with starting line references in the revised manuscript included in square brackets.*

As a preamble I must say that the double standard about code availability in GMD is increasingly frustrating. Either GMD is a journal whose articles describe open-access models or it is not. I do not see the rationale for making an exception for articles describing models that are not open-access for institutional reasons (these definitely fall in the category of non-open-access models). This said the availability of the code of the EQSAM box model version 12 is welcomed.

This manuscript describes the improvements made to the representation of atmospheric aerosols in the ECMWF IFS-COMPO cycle 49R1 in comparison to previous cycles and with observations. In particular this new cycle includes the coupling to the EQSAM4Clim aerosol module for gas-particulate and a calculation of aerosol and precipitation pH.

Overall the manuscript is informative and brings useful information to potential users of the Copernicus aerosol products. Nevertheless the manuscript in its current form suffers from a few weaknesses:

1/ The model description is essentially qualitative (in that it lists the parametrization that have been assembled together) but lacks a more mathematical description of the parametrizations used and the associated numerical schemes. Anything that would provide further mathematical and algorithmic details would be welcomed in the revised manuscript.

*The mathematical and detailed description of EQSAM4Clim is provided in the companion paper GMD-2930 "A revised parameterization for aerosol, cloud and precipitation pH for use in chemical forecasting systems (EQSAM4Clim-v12)". The other updates of cycle 49R1 are described in less detail, as they will be included in the official cycle 49R1 documentation that will be released when cycle 49R1 will be operationally implemented.*

2/ The language and presentation of the manuscript need to be greatly improved. There are many small issues to be fixed, including with tables and figures. The grammar needs to thoroughly checked. Regarding the figures, the labels are too small. The captions often lack details. And the color scales are inappropriate and difficult to read. Some of these issues should have been fixed before publication in EGUsphere.

*Thanks for your suggestion. We checked and rewrote large parts of the manuscripts; most of the Figures have been redrawn, also using color-blind friendly color scales, and a few have been added. We hope that the revised manuscript improves on all of these counts.*

3/ The novelty of this manuscript is to predict aerosol and precipitation pH. The pH of atmospheric aerosols is hardly measurable with state-of-the-art instrumentation so a comparison to direct observations is not possible. A comparison of the IFS model output with indirect observations shows a mixed bag but mostly an acidic bias. In contrast there are many measurements of precipitation pH. The authors focus on only three networks in US, Europe and South-East Asia but ignore other sources of data in Africa, India and some remote locations (e.g. Amsterdam Island). I strongly encourage the authors to look at data from the International Network to study Deposition and Atmospheric chemistry in AFrica (INDAAF) that has long-term measurements of precipitation pH (see https://indaaf.obs-mip.fr/measurements/precipitation/) and could potentially be very useful for evaluating the model performance. It is well known that precipitation pH is often not so acidic in regions with significant emission of soil particles. Precipitation pH over India is known to be not very acidic (doi: 10.4209/aaqr.2015.06.0423) or even alkaline due to crustal material neutralizing the acidity (doi: 10.1016/0004-6981(89)90476-9). Precipitation pH is also not very acidic in some places in Australia (doi: 10.1007/BF01056198). Therefore it is a bit surprising to see the most acidic precipitation over continents in desertic regions on Fig 13a. Further discussion of the model biases on acidity would be highly welcomed. The model captures some features of precipitation pH (eg the east-west gradient in North America) but still has many shortcomings.

*We agree the results clearly could be better. This is however a first implementation and we must leave some room for improvement in the future ;-) We used the three most used networks for this study as they are the most easily accessible, although we are aware of other available networks. This manuscript doesn't have as an objective a full and in-depth evaluation of*

*precipitation pH as estimated by the IFS – this topic would be worth a full manuscript in itself, while in this manuscript this is only a subsection. This is why we didn't expand too much on the evaluation and didn't want to include all networks available. We hope to do this in a future manuscript that will focus solely or mostly on the evaluation of precipitation pH as simulated by the IFS.*

**Other major comments**

Title: I am not sure this is the best title for the article that is broader than just about aerosol acidity.

*Thanks for the suggestion; this comment also meets a comment from reviewer #1. The title has been changed.*

The language of the abstract needs to be more accurate (see minor comments). I strongly encourage the authors to wordsmith the abstract and also discuss the biases in aerosol and precipitation pH.

*Thanks for the suggestion, the abstract has been largely rewritten.*

Bibliography is a little shallow. Here are a few articles that would certainly deserve a citation: doi: 10.1021/acs.accounts.0c00303, 10.1038/s43247-021-00164-0, 10.1021/acs.jpca.8b10676 but I am sure there must be many other relevant papers.

*Thanks a lot for the suggestions, these articles have been cited.*

The sentence "The code revisions that are integrated into the operational version of IFS-COMPO must satisfy the two conditions (one qualitative, one quantitative) that they bring the model closer to "physical" reality, i.e. that more processes and/or species are represented, and that they improve the skill scores against 40 observations" appears to ignore the fact that a particular model development may increase the physical consistency of a model but deteriorate the skill scores if the previous model version relied on error compensation. This is why there is a paradigm shift that consists in retuning the model physics every time a new code revision is made so that a particular model development is given a "fairer" chance to improve the skill scores. I see the current practice as a weakness of the model development process at ECMWF and in NWP centres in general.

*Yes, unfortunately it happens rather often that an improvement of the physical consistency of a parameterization or model component is not associated with an improvement of the skill, or can even provoke a degradation. This can indeed be caused by error compensation, or other problems. In general, in the development of IFS-COMPO, we try to bundle developments together for each cycle, so that the sum of each development improves the model skill more than each development taken separately. The fact that IFS-COMPO is used for an operational use and for a wide community of users also imposes on us the non degradation of skill scores at each model update.*

The terms "rain pH" and "precipitation pH" seem to be used interchangeably. Can the authors clarify if they consider rain only (liquid precipitation) or precipitation (liquid and solid)? In

some places cloud pH is also mentioned but no result is presented. Is cloud pH in scope or not for this manuscript?

*Sorry for the confusion, we consider only precipitation pH (liquid and solid), and we checked the manuscript to remove mentions of "rain pH". Cloud pH is simulated by IFS-COMPO but we chose not to present results, as little observations are available outside of the dataset collected in Pye et al. (2020) and the manuscript has already a fair number of Figures.*

**Minor comments:**

Lines 2, 40, 45, 76 & 568: paper => study

*Corrected, thank you.*

Line 4: as nitrate and ammonium are not in the gas phase per se, the sentence needs modifying.

*Modified, thank you.*

Line 5: "THE global scale"

*Corrected, thank you.*

Line 5: What matters is whether aerosol acidity affects tropospheric chemistry *in the model*. Does it?

Yes – the sentence has been reworded into

*"This information on aerosol acidity influences the simulated tropospheric chemistry processes associated with aqueous phase chemistry and wet deposition."*

Line 9: nitrate and ammonium *in the particulate phase*

*Corrected, thank you.*

Line 16: brought => induced ? It is unclear what are the two quantities which are compared in this sentence.

*Yes, thanks for the suggestion. What is meant in this sentence is that the use of aerosol acidity has a relatively small impact on sulfate production whereas a large impact of the use of EQSAM4Clim on gas/particle partitioning is noted.*

Lines 42 and 45: acidity => aerosol acidity ? aerosol, cloud and precipitation acidity ?

*Corrected, thank you.*

Lines 74-75: this is also true of phosphorus deposition (doi: 10.1038/ncomms3934 and 10.1111/gcb.13766).

*The end of this sentence has been modified to mention this.*

Line 84: delete "input" ?

*Corrected, thank you.*

Lines 84-85: The International Network to study Deposition and Atmospheric chemistry in AFrica (INDAAF) has long-term measurements of precipitation pH (see https://indaaf.obs-mip.fr/measurements/precipitation/) that could potentially be very useful for evaluating the model performance.

*This is a very valuable comment, thank you for this. However, we don't intend this study to be an exhaustive evaluation of IFS-COMPO and of simulated precipitation pH. But you can be sure that we will use this observational dataset for evaluation purposes.*

Line 94: "with three bins for each of these two species"

*Modified, thank you.*

Line 116: consists => consist

*Corrected, thank you.*

Lines 123-125: "CY45R1 and earlier IFS cycles", "provided to the aerosol scheme"

*This part has been removed, as this section focuses on describing CY48R1 IFS-COMPO.*

Line 137: lagrangian => Lagrangian

*Corrected, thank you.*

Line 156: T being a symbol for temperature, it should italicized.

*Done*

Figure 1: please indicate the ion valences on the figure.

*Done.*

Line 172 & 188: dependant => dependent

*Corrected, thank you.*

Line 190: what do you mean by "domain-dependent" ? IFS is a global model so there is no simulation domain in the usual sense of the term. Maybe the authors mean "regionally-dependent" ?

*Chemical domain is meant here : "chemical" has been added in order to avoid confusion.*

Lines 194-195: "… contributions … are … "

This sentence has been rewritten as :

*"The cloud and precipitation pH for each grid box are computed as the cumulative contribution of aerosol acidity, as computed by EQSAM4Clim, and (…)"*

Line 213: " … and smaller than …"

*Corrected, thank you.*

Line 220: "but are both including" => "but both include"

*Corrected, thank you.*

Line 221, "which is used operationally": isn't all of IFS-COMPO used operationally?

*No, some IFS-COMPO parameterizations are not used operationally, and experimental branches exists that are developed for specific purposes, such as dust mineralogy, etc.*

Line 225: convection scavenges chemical species but also transports them upwards and the two processes are intrinsically coupled to each other. How is this dealt with in IFS-COMPO?

*As of now, these two processes (convective vertical transport and scavenging) are treated separately, but are consistent as the outputs of the convection scheme are used in the scavenging scheme when called for convective precipitation. It is planned at some point to indeed treat the scavenging of aerosol/chemistry tracers in the convection scheme, but this has not been done yet.*

Line 226: what is meant by "precipitation fraction"? Is it the fraction of the grid-box where precipitation occurs? How does this vary with model resolution?

*Yes, this is exactly that, as it is not provided by the convection scheme. This value doesn't have a large impact, but it is true that is resolution dependent, and should be revised if a significant update of the horizontal resolution occurs.*

Line 231, "mixed clouds, ie for temperature below the freezing point": this is a weird sentence as it is well known that 1/ liquid clouds can persist below freezing point and 2/ mixed clouds do not occur below say -40°C.

*The sentence has been completed to mention the -40°C threshold.*

Line 232: subjected => subject

*Corrected, thank you.*

Line 235: "… to follow more closely the particle size dependency of Croft et al … This involved …"

*This is much better, thank you!*

Line 241: what does "a measure" mean in this context?

*This is not useful indeed and has been removed from this sentence.*

Line 261: "… aging … slower than … lifetime": a process can be slower than another process but it cannot be slower than a lifetime. A lifetime can be smaller than another lifetime.

*This subsection has been removed as the new aging parameterization is not used operationally in cycle 49R1.*

Section 2.3.3: the reader is missing equations here and what are the many areas of positive results.

*We added the equations of the Gong03 sea-salt aerosol emissions scheme, as a function of whitecap fraction. The main advantage here is to use a parameterization that is applicable to the defined sea-salt aerosol bins, which was not the case of the Monahan et al. (1986) one.*

Line 280: it may be worth saying PM is a concentration, hence the multiplication by air density. It should be said that rho and the aerosol mixing ratios [] are taken in the surface layer.

*This has been added, thank you.*

Line 315: remove dot after 1.5, ie "level 1.5 AOD"

*Corrected, thank you.*

Line 317: Angstrom => Ångström, integrated => evaluated

*Corrected, thank you.*

Lines 358-359: bring together the references into a single parenthesis

*Done, thank you.*

Table 1: ion valence is missing.

*Added.*

Line 383: I am surprised you put India and China on a same footing when it comes to surface PM. Aren't surface aerosol concentrations much larger over India than China, even back in 2019?

*Well, in our simulations they are quite close in terms of yearly average.*

Table 2: please specify the wavelength for the AOD.

*Done, thank you.*

Line 387: an increases => an increase

*Corrected, thank you.*

Lines 419 & 476: c.f. has a specific meaning in English => change to "see Fig. 3" or "see Table 1"

*Corrected, thank you.*

Line 425: close parenthesis

*Corrected, thank you.*

Lines 435 & 445: table 2 => Table 2

*Corrected, thank you.*

Line 490: delete first occurrence of "both"

*Deleted*

Table 3, caption: delete leading "A"

*Deleted*

Line 505: concentration => concentrations

*Corrected, thank you.*

Line 511: is => are

*Corrected, thank you.*

Line 515: not well phrased

*Rephrased as "The use of EQSAM4Clim has the potential to provide an improved regional representation of aerosol, cloud and precipitation pH, which can act as a potential product provided by CAMS"*

Line 516: is => are

*Corrected, thank you.*

Lines 524-525: "using ISORROPIA … and …" w

*Corrected, thank you.*

Line 528: the web link is not effective (at least it didn't respond when I tried)

*Yes, the link is not working anymore; it has been removed.*

Line 532: as mentioned above, there is also a measurement network in Africa that would be very complementary to those used for this evaluation.

*Indeed, thanks again. We'll definitely use this data, but in this study we'd rather stick to the usual China/EU/US networks.*

Line 552: "shown in Fig. 14", word Fig is missing.

*Added, thank you.*

Line 565: are in closer => are closer

*Modified, thank you.*

Line 589: doesn't => does not

*Corrected, thank you.*

Please review the bibliography, eg lines 624 & 693 (remove capitals that are not necessary), line 633 (last author), line 645 (ion valence as exponent)

*Some corrections have been applied to the bibliography.*

Figures: many of the labels are too small. Please enlarge axis legends, labels and ticklabels.

*Most of the Figures have been replotted.*

Figs. 2, 3, 5, 6, 12, 13: the bluish-greenish-greenish-yellowish color scale is particularly difficult to read. I doubt as well that it is legible by blind color readers. The authors are advised to redraw all plots with a better color scale.

*Most of the Figures have been redrawn using different colorscales, which is legible for color blind readers.*

Figs. 2, 3 & 11: please repeat the unit in the caption as it is hard to read from the figures themselves.

*Done.*

Fig. 4: please specify unit.

*Done.*

Figs. 9 and 10: please specify in the caption what the different panels are.

*This has been added, thank you.*

Fig 14c: please add the 1:1 line.

*Done.*

---

## Author Response (AR2)

**Answers to second review of GMD-3072**

Dear Editor,

Thank you for your detailed review of our article, and for the many corrections that you suggest. We corrected all the items that you found and adopted all of your suggestions ; please refer to the tracked-change file for the list of modifications.

Kind regards,

Samuel Remy

---

## Editor Decision (ED2)

[revised manuscript text omitted]
 MB | **0.41** / 0.49 / 0.65 | -0.51 / -0.42 / **-0.22** | **0.43** / 0.75 / 1.36 |
| | RMSE | 1.23 / **1.19** / 1.30 | 1.53 / 1.40 / **1.39** | **2.14** / 2.48 / 3.4 |
| | R | 0.52 / 0.52 / **0.54** | 0.21 / 0.41 / **0.45** | **0.69** / 0.66 / 0.66 |
| $HNO_3$ | mean bias MB | -0.11 / 0.62 / **-0.07** | -0.12 / 0.36 / **-0.10** | |
| | RMSE | **0.50** / 0.87 / 0.50 | **0.50** / 0.91 / 0.65 | |
| | R | 0.33 / **0.57** / 0.30 | **0.21** / 0.20 / 0.20 | |
| $NO_3^-$ | mean bias MB | 1.78 / **0.11** / 1.51 | 1.17 / **-0.08** / 0.6 | 3.19 / **-0.18** / 2.2 |
| | RMSE | 2.2 / **0.74** / 1.97 | 3.55 / **2.71** / 3.02 | 4.25 / **1.47** / 3.3 |
| | R | 0.31 / **0.65** / 0.24 | 0.19 / **0.32** / 0.21 | **0.59** / 0.5 / 0.58 |
| $NH_3$ | mean bias MB | -0.35 / **-0.11** / -0.33 | 0.87 / **0.82** / 0.82 | |
| | RMSE | 1.77 / **1.72** / 1.76 | 1.84 / **1.67** / 1.79 | |
| | R | 0.44 / **0.46** / 0.44 | 0.45 / **0.57** / 0.48 | |
| $NH_4^+$ | mean bias MB | 0.66 / **0.19** / 0.69 | 0.27 / **-0.05** / 0.33 | 1.16 / **0.63** /1.42 |

[revised manuscript text omitted]